

**Herbicide weed control increases nutrient leaching as compared to**
**mechanical weeding in a large-scale oil palm plantation**
Greta Formaglio[1], Edzo Veldkamp[1], Xiaohong Duan [2], Aiyen Tjoa[3], Marife D. Corre[1]
[1]Soil Science of Tropical and Subtropical Ecosystems, University of Göttingen, Göttingen,
37073, Germany
[2]Institute of Biochemical Plant Pathology, Helmholtz Zentrum Munich, 85764, Germany
[3]Faculty of Agriculture, Tadulako University, Palu, 94118, Indonesia
*Correspondence to:* Greta Formaglio (gformag@gwdg.de)



**Abstract**
Nutrient leaching in intensively managed oil palm plantations can diminish soil fertility and
water quality. There is a need to reduce this environmental footprint without sacrificing yield.
We quantified nutrient leaching in a large-scale oil palm plantation on Acrisol soil with factorial
treatment combinations of two fertilization rates (260 N, 50 P, 220 K kg ha$^{-1}$ yr$^{-1}$ as conventional
practice, and 136 N, 17 P, 187 K kg ha$^{-1}$ yr$^{-1}$, equal to harvest export, as reduced management)
and two weeding methods (conventional herbicide, and mechanical weeding as reduced
management). Each of the four treatment combinations was represented by a 2500 m$^2$ plot,
replicated in four blocks. In each plot, soil-pore water was collected monthly at 1.5 m depth for
one year in three management zones: palm circle, inter-row, and frond-stacked area. In the palm
circle, nutrient leaching was low due to low solute concentrations and small drainage fluxes,
resulting from large plant uptake. Conversely, in the inter-row, nitrate and aluminum leaching
losses were high due to their high concentrations, large drainage fluxes, low plant uptake, and
acidic pH. In the frond-stacked area, base cation leaching was high, presumably from frond
litter decomposition, but N leaching was low. Mechanical weeding, even with conventional
high fertilization rates, reduced leaching losses of all nutrients. Mechanical weeding with
reduced fertilization had the lowest N and base cation leaching whereas its yield and economic
gross margin remain comparable with the conventional management practices. Herbicide weed
control decreased ground vegetation, and thereby reduced efficiency of soil nutrient retention.
Our findings signified that mechanical weeding and reduced fertilization should be included in
the Indonesian Ministry of Agriculture program for precision farming (e.g. variable rates with
plantation age), particularly for large-scale plantations, and in the science-based policy
recommendations, such as those endorsed by the Roundtable for Sustainable Palm Oil
association.



## 1 Introduction

Agricultural expansion is a major driver of tropical deforestation (Geist and Lambin, 2002), which have global impacts on reducing carbon sequestration (Asner et al., 2010; van Straaten et al., 2015), greenhouse gas regulation (e.g. Meijide et al., 2020; Murdiyarso et al., 2010), and biodiversity (e.g Clough et al., 2016) and increasing profit gains at the expense of ecosystem multifunctionality (Grass et al., 2020). Oil palm is the most important rapidly expanding tree-cash crop that replaces tropical forest in Southeast Asia (Gibbs et al., 2010; Carlson et al., 2013) due to its high yield with low production costs and rising global demand (Carter et al., 2007; Corley, 2009). Currently, Indonesia produces 57 % of palm oil worldwide (FAO, 2018) and this production is projected to expand in the future, threatening the remaining tropical forest (Vijay et al., 2016; Pirker et al., 2016). Forest to oil palm conversion is associated with a decrease in soil fertility, because of high nutrient export via harvest, reduced rates of soil-N cycling, and decreases in soil organic carbon (SOC) and nutrient stocks (Allen et al., 2015; Allen et al., 2016; van Straaten et al., 2015). The decline in soil fertility reinforces the dependency on fertilizer inputs, and a severe decline can lead to abandonment of the area with further expansion of oil palm plantations in another, exacerbating land-use change. Leaching can contribute to the impoverishment of soil nutrients as well as reduction in water quality and eutrophication of water bodies. Increased nutrient loads to water bodies due to agricultural expansion and intensification, common in temperate areas (Carpenter et al., 1998), are increasingly reported for tropical regions (Figueiredo et al., 2010; Teklu et al., 2018). Given the typically high precipitation rates, leaching losses can possibly be large in intensively managed plantations in the tropics, although deeply weathered tropical soils also have the capacity to store large quantities of N and P (Jankowski et al., 2018; Neill et al., 2013). Indeed, $NO_3^-$, the most leachable form of N, can be retained in the subsoil by anion exchange capacity of highly weathered acidic soils (Wong et al., 1990) whereas P can be fixed to Fe and Al (hydr)oxides of tropical soils (Roy et al., 2016). Nevertheless, there are some evidences of streamwater quality





reductions due to oil palm cultivation in Malaysia (Luke et al., 2017; Tokuchi et al., 2019),
signifying the importance of quantifying nutrient leaching losses in other areas with expansive
oil palm plantations, especially in Jambi, Indonesia, one of the hotspots of forest conversion to
oil palm in Indonesia (Drescher et al., 2016).

Although oil palm plantations can possibly have low leaching losses, as a consequence

of high evapotranspiration and thus low drainage fluxes (Tarigan et al., 2020), most of oil palm
plantations are large-scale enterprises that are characterized by intensive management with high
fertilization rates and herbicide application. Intensive agriculture in the tropics is associated
with high N leaching losses (Huddell et al., 2020). Even in tree-cash or perennial crop
plantations, despite their generally higher evapotranspiration and deeper rooting depth than
annual crops, high fertilization rates result in sustained, large nutrient leaching losses (e.g.
Cannavo et al., 2013; Wakelin et al., 2011). Large $NO_3^-$ leaching from high N fertilization is
always accompanied by leaching of cations (Cusack et al., 2009; Dubos et al., 2017),
impoverishing highly weathered tropical soils that are inherently low in base cations (Allen et
al., 2016; Kurniawan et al., 2018). Fertilization is necessary to support high yields of oil palm
plantations, but reduction in fertilization rates, e.g. to levels that compensate for nutrient export
through harvest, may reduce nutrient leaching losses while maintaining high productivity. On
the other hand, the use of herbicide for weed control can exacerbate nutrient leaching losses, as
prolonged absence of ground vegetation reduces uptake of redistributed nutrients from applied
fertilizers far from reach of crop roots (Abdalla et al., 2019). Herbicide weeding, common in
large-scale oil palm plantation, is practiced in the area where the fertilizers are applied, to reduce
competition for nutrients and water with ground vegetation, and in the inter-rows, to facilitate
access during harvest (Corley and Tinker, 2016). However, herbicide not only eradicates
aboveground vegetative parts but also removes roots slowing down regeneration. In contrast,



mechanical weeding only removes aboveground part, allowing relatively fast regeneration of
ground vegetation, which could take up redistributed nutrients and could reduce leaching losses.
To investigate nutrient leaching losses in an oil palm plantation, the spatial structure
created by the planting design and by the management practices must be taken into account,
which is only partly considered in the sampling designs of previous studies. Three management
zones in oil palm plantations can be identified: (1) the palm circle, an area around the palm´s
trunk where the fertilizers are applied and weeded; (2) the inter-row, weeded less frequently
than the palm circle but unfertilized; and (3) the frond-stacked area, usually every second inter-
row, where the cut senesced fronds are piled up. In these management zones, the interplay of
water fluxes, root uptake and soil nutrient contents determine the extent of nutrient leaching
losses. The palm circle despite having direct fertilization have also large water and nutrient
uptake (Nelson et al., 2006) because of high root density (Lamade et al., 1996) such that large
leaching losses may only occur following pulse high fertilization and during high drainage
(from high precipitation) events (Banabas et al., 2008a). The inter-row experiences higher water
input from precipitation than the palm circle because of lower canopy interception (Banabas et
al., 2008b), and large water flux within the soil because of low root uptake, stimulating nutrient
transport to lower depths. However, as there is no direct fertilizer application on the inter-row,
nutrient leaching may be low. The frond-stacked area receives nutrients from decomposition of
nutrient-rich fronds (Kotowska et al., 2016) and such mulching with senesced fronds prevents
runoff and promotes water infiltration as a consequence of enhanced macroporosity by
increased organic matter (Moradi et al., 2015). High water infiltration may generate high water
drainage fluxes, resulting in intermediate nutrient leaching losses in the frond-stacked area.
In this study, we aimed to quantify nutrient leaching losses in an intensively managed,
large-scale oil palm plantation, and to assess if reduced intensity of management (i.e. reduced
fertilization rates equal to harvest export and mechanical weeding) can reduce leaching losses



in oil palm plantations. We tested these hypotheses: (1) leaching losses in the palm circle will
be larger than in the other management zones because of direct fertilizer application; (2)
leaching losses under herbicide application will be higher than mechanical weeding because of
slower regeneration of ground vegetation that can augment nutrient retention; (3) nutrient
leaching fluxes under conventional high fertilization rates will be substantial compared to
reduced rates because of excessive nutrient inputs. Our study provides a systematic
quantification of an important environmental footprint of oil palm production, taking into
consideration its spatial variation in management zones, and evaluates the effectiveness of
alternative management practices for leaching reduction.
**2 Materials and methods**
**2.1 Study area and experimental design**
This study was conducted in a state-owned oil palm plantation in Jambi province, Indonesia (1°
43' 8" S, 103° 23' 53" E, 73 m above sea level). Mean annual air temperature is 26.7 ± 1.0 °C
and mean annual precipitation is 2235 ± 385 mm (1991–2011; data from Sultan Thaha airport,
Jambi). During our study period (March 2017–February 2018), the mean daily air temperature
was 26.3 °C and annual precipitation was 2772 mm, with a dry period between July and October
(precipitation $< 140$ mm month$^{-1}$). The soil is highly weathered, loam Acrisol soil (Allen et al.,
2015) and nutrient inputs from bulk precipitation in the area, measured in 2013, were 12.9 kg
N, 0.4 kg P, 5.5 kg K ha$^{-1}$ yr$^{-1}$ (Kurniawan et al., 2018).
This oil palm plantation was established between 1998 and 2002, and so the palms were
16–20 years old during our study period. The plantation encompassed 2025 ha, with a planting
density of approximately 142 palms ha$^{-1}$, spaced 8 m apart on rows. The rows between palms
are used alternately for harvesting operations and to pile-up senesced fronds, which are
regularly cut to facilitate harvesting of fruits; this frond-stacked area covers 15 % of the



plantation. The palm circle, 2 m radius from the trunk, wherein fertilizers are applied and
weeded four times a year, covers 18 % of the plantation. The remaining 67 % can be classified
as inter-row, which is not fertilized but weeded two times a year.
In November 2016, a two (fertilization rates) by two (weeding methods) factorial
management experiment was established in this plantation as part of the framework of the
EFForTS project, described in detail by Darras et al. (2019). For fertilization treatments, the
conventional rates were 260 N, 50 P, 220 K kg ha$^{-1}$ yr$^{-1}$, whereas the reduced rates were 136 N,
17 P, 187 K kg ha$^{-1}$ yr$^{-1}$. Reduced fertilization rates were determined to compensate for nutrient
exports via fruit harvest and were based on the nutrient concentrations measured in the fruit
bunches multiplied by the annual yield. The fertilizer sources were urea ($CH_4N_2O$), triple
superphosphate ($Ca(H_2PO_4)_2 \cdot H_2O$) and muriate of potash ($KCl$), and these were applied
according to the plantation's standard practices: split in two applications per year (in April and
October), applied in a band within a 2 m radius from the palm, and this area was raked before
fertilizer application. For both fertilization treatments, lime (426 kg dolomite ha$^{-1}$ yr$^{-1}$;
$CaMg(CO_3)_2$) and micronutrients (142 kg Micro-Mag ha$^{-1}$ yr$^{-1}$ with 0.5 % $B_2O_3$, 0.5 % $CuO$,
0.25 % $Fe_2O_3$, 0.15 % $ZnO$, 0.1 % $MnO$ and 18 % $MgO$) were also applied besides the N, P
and K fertilizers, as commonly practiced in large-scale plantations on acidic Acrisol soils
(Pahan, 2010). For weeding treatments, the conventional method was the use of herbicide
(glyphosate), whereas the reduced method was mechanical weeding using a brush cutter.
Glyphosate was applied following plantation's standard practice: 1.5 L ha$^{-1}$ yr$^{-1}$ to the palm
circle, split four times a year, and 0.75 L ha$^{-1}$ yr$^{-1}$ to the inter-row, split two times a year. The
mechanical weeding was carried out in the same areas and frequencies as herbicide application.
This management experiment comprised of four replicate blocks and each had four plots (50 m
x 50 m each) assigned to four treatment combinations: conventional rate–herbicide,





conventional rate–mechanical weeding, reduced rate–herbicide, and reduced rate–mechanical
weeding.

### 2.2 Soil water sampling

We collected monthly soil-pore water samples over one year, using suction cup lysimeters (P80
ceramic, maximum pore size 1 μm; CeramTec AG, Marktredwitz, Germany). We installed the
lysimeters in January 2017, choosing two palms per plot and sampling in the three management
zones: 1) in the palm circle, at 1 m from the palm trunk, 2) in the frond-stacked area, at about
4 m from the palm trunk, and 3) in the inter-row, at approximately 4 m from the palm trunk
(Fig. A1). In total, 96 lysimeters were installed (4 treatment plots x 4 replicates x 2 subplots x
3 management zones). The lysimeters were inserted into the soil till 1.5 m depth, so that the
soil-pore water was collected well below the rooting depth of 1 m which is common to oil palm
plantations on loam Acrisol soils near our study site (Kurniawan et al., 2018). Starting in March
2017, soil water was sampled by applying 40 kPa vacuum (Kurniawan et al., 2018; Dechert et
al., 2005) to the lysimeters and collected in dark glass bottles, which were stored in a bucket
buried in the field. Once a week, we transferred the collected water into plastic bottles and
transported them to the field station, where they were stored frozen. The collection continued
over a month until a volume of 100 mL was collected from each lysimeter, or until the end of
the month. The frozen water samples were transported by air freight to the University of
Goettingen, Germany, where element concentrations were determined. We measured the
concentrations of mineral N ($NH_4^+$ and $NO_3^-$), total dissolved N (TDN) and Cl by continuous
flow injection colorimetry (SEAL Analytical AA3, SEAL Analytical GmbH, Norderstadt,
Germany), as described in details by Kurniawan et al. (Kurniawan et al., 2018). Dissolved
organic N (DON) was calculated as the difference between TDN and mineral N. We measured
the concentrations of base cations (Na, K, Ca, Mg), total Al, total Fe, total Mn, total S, and total



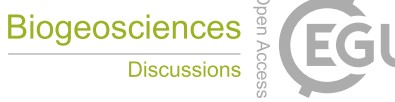

P with an inductively coupled plasma–atomic emission spectrometer (iCAP 6300; Thermo
Fischer Scientific GmbH, Dreieich, Germany).

We determined a partial cation-anion charge balance of the major elements

(concentrations > 0.03 mg L$^{-1}$) in soil-pore water by converting the concentrations to $\mu mol_{charge}$
L$^{-1}$. We assumed S to be in the form of sulfate (SO$_4^{2-}$) and total Al to have a charge of 3$^+$. We
calculated the contribution of organic acids (RCOO$^-$) and bicarbonate (HCO$_3^-$) as the difference
between the measured cations and anions (2018).

**2.3 Modeling water drainage**
The water balance was modeled using the water sub-model of the Expert-N software, version
5.0 (Priesack, 2005), which was successfully used to estimate drainage fluxes from different
land uses in Indonesia (Dechert et al., 2005; Kurniawan et al., 2018). The model inputs were
climate data (solar radiation, temperature, precipitation, relative humidity, and wind speed), and
soil (texture, bulk density, and hydraulic functions) and vegetation characteristics (biomass,
leaf area index, and root distribution). The climate data were taken from the climatological
station in the plantation (described in detail by Meijide et al., 2017), and the oil palm biomass
was taken from a study on oil palm plantations near our study site (Kotowska et al., 2015). Soil
bulk density and porosity in the top 10 cm were measured in each management zone at our
study site, whereas for the 10–50 cm depth these were measured in the inter-row, assuming that
the differences in soil bulk density among management zones would be minimal below the
topsoil. Data for soil bulk density and porosity for the 50–200 cm depth, as well as soil texture,
soil hydraulic parameters (i.e. water retention curve, saturated hydraulic conductivity and Van
Genuchten parameters for the water retention curve), and root distribution were taken from
Allen et al. (2015) and Kurniawan et al. (2018), choosing their studied oil palm plantations





closest to our study site. Expert-N water sub-model calculates daily water drainage based on
precipitation, evapotranspiration, canopy interception, runoff, and change in soil water storage.
Evapotranspiration is calculated using Penman-Monteith method (Allen, 1998), applying a
plant factor of 1.06 (Meijide et al., 2017), with plant transpiration based on leaf area index
(LAI), plant biomass, and maximum rooting depth. The canopy interception is calculated from
the percentage of throughfall and the maximum water storage capacity of the canopy. Runoff
is calculated from soil texture and bulk density, which determine the water infiltration rate, and
from the slope, which was 5 % (Röll et al., 2019). The vertical water movement is calculated
using Richard´s equation based on soil hydraulic functions.

To model the drainage in the different management zones, we used the measured soil

bulk density and porosity in the top 10 cm and adjusted other input parameters to simulate
differences in water balance in each management zone. For the palm circle, we set the LAI to
3.65, which is the maximum LAI measured at our site (Fan et al., 2015), to simulate high water
uptake in the palm circle (Nelson et al., 2006) and maximum rooting depth to 1 m, which is
reported for oil palm plantations near our site (Kurniawan et al., 2018). The percentage
throughfall in the palm circle was set to 50 % and the water storage capacity of oil palm trunk
was set to 8.4 mm (Tarigan et al., 2018). For the inter-row, we set the LAI and the maximum
rooting depth as half of the palm circle (1.8 LAI, 50 cm rooting depth), as roots are shallower
between palms (Nelson et al., 2006); the throughfall was set to 10 %, and the palm trunk's water
storage capacity was set to 4.7 mm (based on canopy storage capacity reported by Tarigan et
al., 2018). For the frond-stacked area, the LAI was set to 0.75, which is half of the minimum
measured in the studied plantation (Darras et al., 2019), as understory vegetation is absent at
this zone. Values for interception in the frond-stacked area was set to the same values as the
inter-row, whereas the runoff was set to 0, as mulching with senesced fronds slows down runoff
(Tarigan et al., 2016).



For validation of the Expert-N water sub-model outputs, we measured soil water matric
potential at depths of 30 cm and 60 cm over the study period and compared the measured values
with the modeled matric potential. Matric potential was measured by installing a tensiometer
(P80 ceramic, maximum pore size 1μm; CeramTec AG, Marktredwitz, Germany) at each depth
in each management zone near to two palms in two treatments (i.e. conventional rate–herbicide,
and reduced rate–mechanical weeding), for a total of 12 tensiometers. We summed the modeled
daily drainage at 1.5 m depth to get the monthly drainage fluxes, which we then multiplied with
the element concentrations in soil water to get the monthly nutrient leaching fluxes.

**2.4 Soil biochemical characteristics and nutrient retention efficiency**

We measured soil biochemical properties in the same sampling locations (Figure S1) at four
depth intervals: 0–5 cm, 5–10 cm, 10–30 cm, and 30–50 cm. Soil samples from the same
management zone in each plot were pooled to make one composite sample, totaling to 192 soil
samples (4 treatments plots x 4 replicates x 3 management zones x 4 depths). The samples were
air-dried and sieved (2 mm) and measured for pH (1:4 soil-to-water ratio) and for effective
cation exchange capacity (ECEC), by percolating the soils with unbuffered 1 mol $L^{-1}$ $NH_4Cl$
and measuring the cations (Ca, Mg, K, Na, Al, Fe, Mn) in percolates using ICP-AES. A
subsample was finely ground and analyzed for organic C and total N using a CN analyzer (Vario
EL Cube, Elementar Analysis Systems GmbH, Hanau, Germany), and for $^{15}N$ natural
abundance signature using isotope ratio mass spectrometer (IRMS; Delta Plus, Finnigan MAT,
Bremen, Germany). We calculated the soil element stocks for each depth by multiplying the
element concentration with the measured bulk density and summed for the top 50 cm; other soil
characteristics (e.g. pH, ECEC, base saturation) in the top 50 cm soil were calculated as the
depth-weighted average of the sampled depths.



258 In addition, we calculated the N and base cation retention efficiency in the soil for each

259 experimental treatment and management zone following the formula: nutrient retention

260 efficiency = 1 – (nutrient leaching loss / soil-available nutrient) (Kurniawan et al., 2018). We

261 used the gross N mineralization rates in the top 5 cm soil (Table A1) as an index of soil-available

262 N whereas soil-available base cations was the sum of the stocks of K, Na, Mg and Ca in the top

263 10 cm soil, expressed in $mol_{charge}$ m$^{-2}$.

265 **2.5 Statistical analyses**

266 For soil biochemical properties measured once, we tested for differences among management

267 zones as well as among experimental treatments for the entire 50 cm depth, using the analysis

268 of variances (ANOVA) with Tukey HSD as a post hoc test. The soil variables that showed non-

269 normal distribution or unequal variances, tested with Shapiro–Wilk and Levene's tests,

270 respectively, were log-transformed prior to the analysis. Base cation and N retention efficiency

271 were also tested for differences between experimental treatments in the same way. For

272 repeatedly measured variables, i.e. soil-pore water solute concentrations and leaching fluxes,

273 we used linear mixed-effects models (LME; Bates et al., 2015) to assess the differences among

274 management zones and treatments. For testing management zone differences, we conducted the

275 LME with management zone as fixed effect and random effects for sampling months and

276 experimental treatments nested with replicate plots, which were also nested with subplots. For

277 testing treatment differences, we calculated for each replicate plot on each sampling month the

278 area-weighted average of the three management zones (i.e. palm circle accounts for 18 % of the

279 plantation area, the frond-stacked area 15 %, and the inter-row 67 %), and LME was carried

280 out with treatment as fixed effect and random effects for sampling months and replicate plots

281 nested with subplots. If the residuals of the LME models were not normally distributed, we

282 applied either logarithmic or square root transformation. Differences were assessed with



ANOVA (Kuznetsova et al., 2017) followed by Tukey HSD (Hothorn et al., 2008). We also
used LME to assess differences in soil water matric potential among management zones, with
management zone as fixed effect and measurement days and depth nested with treatment as
random effects. Comparability between modeled and measured soil water matric potential for
each depth in each management zone ($n = 50$ field measurements) was assessed using Pearson
correlation test. All tests were considered significant at $P \leq 0.05$, except for soil pH which we
considered a marginal significance at $P = 0.06$. All statistical analyses were performed with R
version 3.6.1 (R Core Team, 2019).

## 292    3 Results

### 293    3.1 Soil biochemical properties and water balance

Soil biochemical properties in the top 50 cm did not differ between experimental treatments (all
$P > 0.05$) but strongly differed among management zones (Table 1). The frond-stacked area,
where senesced fronds were regularly piled like mulch material, had higher SOC and total N
stocks ($P < 0.01$) compared to the other management zones. The inter-row, with regular
weeding but without direct fertilizer and lime inputs, showed lower exchangeable base cation
contents (i.e. Ca, Mg, K) compared to the other management zones ($P \leq 0.02$) and higher
exchangeable Al content than the palm circle ($P = 0.01$). This was reflected in the lower base
saturation and higher Al saturation in the inter-row compared to the other zones ($P < 0.01$).
Also, inter-row had the lowest ECEC ($P < 0.01$) and marginally lower pH than the palm circle
($P = 0.06$). The palm circle, where fertilizers and lime were applied, had generally comparable
exchangeable element contents with the frond-stacked area, except for K, which was higher in
the palm circle ($P < 0.01$), and for Mn, which was higher in the frond-stacked area ($P < 0.01$).



There were high positive correlations between field-measured and modeled soil water
matric potential (Fig. 1). The matric potential was generally lowest in the palm circle,
intermediate in the inter-row, and highest in the frond-stacked area ($P < 0.01$). This pattern was
also reflected in the low drainage flux in the palm circle and high drainage flux in the frond-
stacked area (Table 2; Fig. 2). In the palm circle, the low drainage flux had resulted from high
plant transpiration and interception whereas the high drainage flux in the frond-stacked area
was due to low evapotranspiration and runoff with the senesced frond mulch (Table 2). In ratio
to annual precipitation, the calculated annual evapotranspiration was 51 %, 31 %, and 38 % in
the palm circle, frond-stacked area, and inter-row, respectively; annual drainage fluxes at 1.5
m depth were 20 % of precipitation in the palm circle, 65 % in the frond-stacked area, and 43
% in the inter-row. Seasonally, the monthly drainage fluxes had two peak periods, May and
November, after consecutive days of moderate rainfall, and were lowest during the end of the
dry season towards the start of the wet season (Fig. 2).

**3.2 Differences in leaching losses among management zones and treatments**

For element concentrations in soil-pore water at 1.5 m depth, treatment differences were
exhibited clearly in the palm circle and inter-row (Fig. 3), with the herbicide treatment showing
higher element concentrations than the mechanical weeding ($P \leq 0.02$). The frond-stacked area
had generally lower ionic charge concentrations compared to the other management zones (Fig.
3). The dominant cations were $Al^{3+}$, $Ca^{2+}$, $Mg^{2+}$, $K^+$, and $Na^+$ across experimental treatments
and management zones. Among the management zones, $Al^{3+}$ concentrations were highest in the
inter-row, intermediate in the palm circle, and lowest in the frond-stacked area ($P < 0.01$). The
concentrations of $Ca^{2+}$ were similar in the palm circle and frond-stacked area ($P = 0.42$), and
these were higher than the inter-row ($P < 0.01$). The concentrations of $Mg^{2+}$ and $K^+$ were higher
in the palm circle than in the other two management zones ($P < 0.01$). The $Na^+$ concentrations





were higher in the palm circle and inter-row than in the frond-stacked area ($P < 0.01$). As for
dissolved N, $NH_4^+$ concentrations were lowest in the frond-stacked area, followed by the palm
circle, and highest in the inter-row ($P = 0.01$). Across treatments, $NH_4^+$ was 4–18 % of TDN
whereas DON was 1-7 % of TDN. Thus, $NO_3^-$ was the main form of dissolved N, and this was
highest in the inter-row, followed by the frond-stacked area, and lowest in the palm circle ($P <$
$0.01$). The dominant anion was $Cl^-$ with higher concentrations in the palm circle than in the
other zones ($P < 0.01$).
Monthly leaching fluxes showed a common pattern among the major solutes (Fig. 4):
there were two peaks of leaching losses (May and November) that followed fertilizer
applications, and lower leaching losses during the dry season from July to October. Leaching
fluxes of $NO_3^-$ showed similar pattern as its concentrations: higher in the inter-row, followed
by the frond-stacked area, and lowest in the palm circle ($P < 0.01$; Fig. 4). Total Al leaching
fluxes were also higher in the inter-row than the other zones ($P < 0.01$; Fig. 4). On the other
hand, although base cation concentrations were large in the palm circle (Fig. 3), the low
drainage fluxes in this zone (Fig. 2; Table 2) resulted in opposite patterns of base cation leaching
fluxes among management zones; Ca, K, and Mg leaching were higher in the frond-stacked area
than the palm circle and inter-row (all $P < 0.01$; Fig. 4). Leaching of Na was higher in both the
frond-stacked area and inter-row than the palm circle ($P < 0.01$; Fig. 4).
Reduced intensity of management clearly influenced nutrient leaching losses (Fig. 5;
Table 3). Specifically, mechanical weeding reduced $NO_3^-$ and cation leaching compared to
herbicide weed control ($P \leq 0.03$; Fig. 5; Table 3). Leaching of $NO_3^-$ was highest in the
conventional fertilization–herbicide treatment and lowest in reduced management treatments
($P \leq 0.02$; Fig. 5). This was also reflected in the leaching fluxes of accompanying cations;
specifically, total Al and Ca leaching were higher in conventional fertilization–herbicide
treatment than the reduced management treatments (all $P \leq 0.02$; Fig. 5). For the other base





cations, mechanical weeding clearly lowered leaching losses compared to herbicide weeding,
in particular K and Na leaching in both fertilization rates and Mg leaching in conventional
fertilization (all $P \leq 0.03$; Fig. 5).

### 3.3 Annual leaching losses and nutrient retention efficiency

In proportion to the applied fertilizer, annual leaching losses of TDN (Table 3) were 28 % of
the applied N in the herbicide treatment for both conventional and reduced fertilization rates,
24 % in the mechanical weeding with conventional fertilization, and only 19 % in the
mechanical weeding with reduced fertilization. The annual leaching of K (Table 3) was 4 % of
the applied K fertilizer in the herbicide treatment and 3 % in the mechanical weeding for both
fertilization rates. In this highly weathered Acrisol soils with high capacity for P fixation by Fe
and Al (hydr)oxides, there was no leaching of dissolved P (Table 3).

Both N and base cation retention efficiencies were generally lower in the inter-row

compared to the other management zones ($P \leq 0.03$), except for reduced fertilization–
mechanical weeding where there were no differences among management zones (Table 4). The
area-weighted average N retention efficiency was comparable among experimental treatments
($P = 0.89$) but there was a trend of increasing efficiency with decreasing management intensity
(Table 4). Base cation retention efficiency showed clear differences among experimental
treatments for each management zones: in the palm circle, it was highest in mechanical weeding
and lowest in the herbicide treatment ($P = 0.04$); in the frond-staked area and inter-row, it was
lowest in the most intensive management treatment (conventional fertilization–herbicide) and
highest in either mechanical weeding or reduced fertilization ($P \leq 0.05$; Table 4). The area-
weighted average base cation retention efficiency was also clearly influenced by weeding





method, being lowest in herbicide treatment and highest in mechanical weeding both with
conventional fertilization ($P = 0.03$; Table 4).

**4 Discussion**
**4.1 Water model and temporal pattern of nutrient leaching losses**
To our knowledge, this study is the first attempt to model drainage fluxes from the different
management zones of an oil palm plantation, making our comparisons with literature values
limited. Our modeled annual transpiration rate in the palm circle (Table 2) was remarkably
similar to the values estimated with the same Penman–Monteith method (827–829 mm yr$^{-1}$;
Meijide et al., 2017; Röll et al., 2019), and our average daily transpiration rate (2.3 mm d$^{-1}$) was
within the range of that measured with drone-based photogrammetry (3 ± 1 mm d$^{-1}$;
Ahongshangbam et al., 2019), all in the same oil palm plantation. Also, the modeled annual
runoff in the palm circle and inter-row (Table 2) was within the range of runoff estimates in oil
palm plantations in Jambi province (10–20 % of rainfall; Tarigan et al., 2016) and in Papua
New Guinea (1.4–6 % of rainfall; Banabas et al., 2008b). Considering the areal proportions of
the three management zones, the weighted-average drainage flux (1161 mm yr$^{-1}$) was lower
than that estimated for smallholder oil palm plantations near our study site (1614 mm drainage
flux with 3418 mm precipitation measured in 2013; Kurniawan et al., 2018), although their
ratios to annual precipitation were comparable. Aside from the difference in precipitation
during our study period compared to the relatively wet year of 2013, evapotranspiration rate is
higher in large-scale than smallholder oil palm plantations in our study area (Röll et al., 2019),
which would lead to lower drainage flux in large-scale plantation. Moreover, in the frond-
stacked area, enhanced porosity from organic matter that facilitates water infiltration (Moradi
et al., 2015), as indirectly indicated by its low soil bulk density (Table 1), combined with low





evapotranspiration and runoff, resulted in large drainage flux (Table 2). This suggests that piling
senesced fronds may amend groundwater recharge, which could moderate discharge
fluctuations in water catchments of oil palm converted areas (Tarigan et al., 2020). Based on
these comparisons with literature values and on the good agreement between modeled and
measured soil water matric potential (Fig. 1), we conclude that our modeled drainage fluxes
were reliable.

The temporal peaks of nutrient leaching fluxes (May and November; Fig. 4) had resulted

from the combined effect of high drainage flux and fertilizer application. The high drainage
fluxes in May and November (Fig. 2) might have stimulated the downward transport of
elements and decreased their residence time in the soil, and thus their adsorption onto the soil
exchange sites (Lohse and Matson, 2005). These high water fluxes usually dilute the element
concentrations in the soil-pore water; however, high concentrations were maintained because
of fertilizer and lime applications in the same periods, resulting in parallel peaks of drainage
and leaching fluxes (Figs. 2 and 4). The high $NO_3^-$ leaching following urea-N fertilization (Fig.
4) suggests increased nitrification (Silver et al., 2005), fast $NO_3^-$ transport through the soil
column, and reduced anion adsorbtion capacity, which otherwise would have delayed anion
leaching (Wong et al., 1990). The latter was possibly aggravated by the additional $Cl^-$ from
fertilization with KCl (Fig. 3), which could saturate the soil anion exchange sites, particularly
at this mature plantation with already 16–20 years of high fertilization rates. Large $NO_3^-$
leaching is always accompanied by large leaching of buffering cations (Dubos et al., 2017;
Kurniawan et al., 2018), resulting in their similar temporal patterns (Fig. 4). These findings
showed that fertilization should be avoided during periods of high drainage fluxes. Generally,
the high drainage was a consequence of a protracted period of moderate rainfall (Fig. 2).
Prediction of periods of high precipitation and drainage will further be confounded by climate
change, which is widening the range between wet and dry seasons and increasing the





uncertainties in rainfall intensity and distribution (Chou et al., 2013; Feng et al., 2013).
Fertilization during the dry period is also not advisable given the high volatilization of applied
urea even in acidic soil as this is always accompanied by liming (Goh et al., 2003; Pardon et
al., 2016) and the low palm uptake during the dry season (Corley and Tinker, 2016). Thus,
reduction of fertilization rates, e.g. at compensatory level equal to harvest export, seems a viable
option to reduce leaching losses without sacrificing production. One other option is the use of
organic amendments and slow-release fertilizers, which have been shown to reduce N leaching
in tropical cropping systems (Nyamangara et al., 2003; Mohanty et al., 2018; Steiner et al.,
2008) and to improve soil fertility in oil palm plantations (Comte et al., 2013; Boafo et al.,
2020), as was also evident with mulching of senesced oil palm fronds (i.e. high SOC, total N,
ECEC and base saturation in the frond-stacked area; Table 1).

### 4.2 Leaching losses in the different management zones

Contrary to our first hypothesis, leaching losses were generally higher in the inter-row,
especially for mineral N (largely $NO_3^-$; Fig. 3), compared to the other zones, whereas the palm
circle had the lowest leaching (Fig. 4). This strikingly large mineral N leaching losses in the
inter-row were surprising given that this area did not receive direct fertilizer inputs (see section
2.1). This result suggests that mineral N was transported from the directly fertilized palm circle
to the inter-row via surface and subsurface lateral flow as these two zones were just 3 m apart
(Fig. A1). Surface transport of mineral N was probably a minor process at our site because of
the low runoff (Table 2); in an oil palm plantation in Papua New Guinea, the loss of N fertilizer
via surface runoff is only 0.3–2.2 kg N ha$^{-1}$ yr$^{-1}$ (Banabas et al., 2008b). Mineral N was probably
predominantly transported to the inter-row via subsurface lateral flow. Acrisol soils are
characterized by clay translocation from upper to lower depths that could create an impeding
layer conducive to lateral water flow (Elsenbeer, 2001). Indeed, the clay contents of the Acrisol





soils at our study area increase with depth, and soil bulk density is highest at 100–150 cm than
at 150–200 cm depth (Allen et al., 2016). In addition, the palm roots spreading from the palm
circle to the inter-row may create channels for subsurface lateral flow of dissolved ions like
$NO_3^-$ (Li and Ghodrati, 1994). Higher mineral N leaching in the inter-row than palm circle was
also observed in Brazil and it was attributed to lower root density and higher N mineralization
at increasing distance from the palm´s trunk (Schroth et al., 2000). Hence, a combination of
lower root uptake, higher N mineralization, and subsurface lateral transport (particularly for
$NO_3^-$) may all have contributed to higher mineral N leaching losses in the inter-row than the
palm circle. The main accompanying cation for $NO_3^-$ leaching in the inter-rows was $Al^{3+}$ (Figs.
3 and 4), as this zone's soil pH (Table 1) was within the Al-buffering range (pH 3–5; van
Breemen et al., 1983), having no direct lime application and thus low base saturation (Table 1).
Our findings showed that if leaching is measured only within the palm circle, this largely
underestimates mineral N and Al leaching losses.

The palm circle had relatively low N leaching losses (Figs. 3 and 4) despite the direct

application of fertilizer. This was probably due to the large root density in this zone that
facilitates an efficient nutrient uptake (Edy et al., 2020; Nelson et al., 2006). Hence, the
dominant anion in soil-pore water in the palm circle was $Cl^-$ (Fig. 3), enhanced by the applied
KCl fertilizer, which was accompanied by high base cation concentrations relative to dissolved
Al (Fig. 3). The former was due to the applied micromag fertilizer and dolomite (section 2.1),
which increased pH and exchangeable bases and rendered Al in insoluble form (i.e. lower
exchangeable Al; Table 1; Schlesinger and Bernhardt, 2013). Despite their high concentrations,
the leaching fluxes of base cations in the palm circle (Fig. 4) were constrained by the low water
drainage flux due to high evapotranspiration (Table 2).

The frond-stacked area was at the same distance from the palm circle as the inter-row

(Fig. A1) but had substantially lower mineral N leaching losses (Figs. 3 and 4). Decomposition





of nutrient-rich fronds (Kotowska et al., 2016) resulted in high SOC and N stocks (Table 1),
which can support large microbial biomass in this zone (Haron et al., 1998). Thus, the low
mineral N leaching in the frond-stacked area may be attributed to immobilization of mineral N
by large microbial biomass, converting mobile $NO_3^-$ to less mobile organic N (e.g. Corre et al.,
2010). In addition, it could be possible that palm root uptake of nutrients (including mineral N)
was higher in the frond-stacked area compared to the inter-row as roots proliferate in nutrient-
rich zones (Table 1; Hodge, 2004). This is supported by studies that showed higher root density
and higher water uptake under the frond piles compared to the inter-row (Rüegg et al., 2019;
Nelson et al., 2006). The high ECEC, base saturation and pH in frond-stacked area (Table 1),
despite having no direct lime application, were due to the release of nutrients from
decomposition of frond litter, which contain high levels of base cations (Kotowska et al., 2016).
Thus, although leaching of base cations were larger in the frond-stacked area than in the inter-
row (Fig. 4), these losses merely mirrored their high exchangeable levels (Table 1). Finally, the
leaching of Al was low in the frond-stacked area (Figs. 3 and 4) because Al becomes insoluble
as pH increased (i.e. lower exchangeable Al; Table 1). Altogether, these results highlighted the
benefits of piling senesced fronds onto the soil to reduce leaching of mineral N and Al, which
otherwise can potentially diminish ground water quality, and to amend soil fertility (Table 1).
Oil palm plantations in other areas (e.g. Borneo; Rahman et al., 2018) were reported to practice
piling of senesced fronds on every inter-row, which we did not observed in our study region as
that is claimed to hinder access to palms during harvest; nonetheless, our findings implied that
increase in the frond-stacked area can contribute to sustainable management practices of oil
palm plantations.

**4.3 Leaching losses under different intensity of management**



There was a clear influence of management intensity treatments on nutrient leaching losses with
a general reduction of leaching in reduced management intensity (Fig. 5; Table 3). In line with
our second hypothesis, the weeding methods clearly influenced leaching losses with a common
pattern of lower leaching fluxes in mechanical weeding than herbicide treatment (Fig. 5; Table
3). Mechanical weeding was associated with more ground vegetation cover (Darras et al., 2019)
and higher nutrient retention efficiency than herbicide weeding (Table 4), suggesting that faster
regrowth of understory vegetation by mechanical weeding have additionally contributed to the
uptake of nutrients and thus reducing leaching losses. This is in line with some studies in
temperate forests and a cedar plantation, which showed that understory vegetation can take up
excess $NO_3^-$ in the soil (Olsson and Falkengren-Grerup, 2003) and reduce $NO_3^-$ leaching and
the mobilization of Ca and Mg (Baba et al., 2011; Fukuzawa et al., 2006). Enhanced understory
vegetation in oil palm plantations may also positively impact biodiversity by increasing plant
species richness and soil macrofauna diversity and abundance (Luke et al., 2019; Ashton-Butt
et al., 2018), which may facilitate uptake and recycling of nutrients. Increase in soil macrofauna
might have contributed to lower leaching of Na with mechanical weeding (Fig. 5), since
herbivores and decomposers take up a large amount of Na (Kaspari et al., 2009). In addition,
the use of glyphosate is associated with possible health risks to workers and the environment
(van Bruggen et al., 2018); also, the economic gross margin (i.e. revenues minus costs) is
comparable between mechanical weeding and herbicide treatment because of needed labor for
periodic mechanical cutting of resistant ground vegetation in oil palm plantations with herbicide
weeding (Darras et al., 2019; Pahan, 2010). Altogether, these results advocate for the higher
sustainability of mechanical weeding over herbicide application.

The reduction of N fertilization rates decreased $NO_3^-$ leaching, supporting our third

hypothesis. Comparing conventional and reduced fertilization rates, there were no differences
in total N stocks (section 3.1), mineral N levels (Darras et al., 2019), N retention efficiency





(Table 4) and oil palm yield (Darras et al., 2019), suggesting that excess N (above harvest
export; section 2.1) from high N fertilization was largely lost through leaching (Table 3). The
decreased Al and Ca leaching with reduced fertilization can be attributed to the lowered $NO_3^-$
leaching, since these were the accompanying cations (Figs. 4 and 5). Also, a reduction of Ca
leaching could have resulted from the lower application rate of triple superphosphate fertilizer,
which contains 16 % of Ca. The reduced K fertilization had no effect on K leaching (Fig. 5)
because K fertilization rate was only reduced by 15 % of the conventional rate due to high K
requirements of oil palm fruits (section 2.1). We conclude that this mature (16–20 years old)
plantation with conventional management was overly fertilized for N, and that a reduction in N
fertilization rate may be included in the Indonesian program for precision farming (Ministry of
Agriculture of Indonesia, 2016) to reduce environmental footprint of oil palm production.

Comparing the N leaching losses in the studied plantation with other fertilized tropical

plantations (Table A2), our plantation had higher N leaching than other large-scale oil palm
plantations on similar soils with comparable fertilization rates (Omoti et al., 1983; Tung et al.,
2009). However, in these studies the leaching losses were measured in the palm circle (Omoti
et al., 1983) or the sampling location was not specified (Tung et al., 2009), such that N leaching
may be underestimated as our results showed the high contribution of the inter-row to leaching
losses (Figs. 3 and 4). The N leaching fluxes in our plantation were also higher than in
smallholder oil palm plantations in the same area, which typically had much lower fertilization
rates (Kurniawan et al., 2018). On the other hand, our plantation had lower N leaching losses
than an oil palm plantation and coffee agroforestry systems on volcanic soils (Banabas et al.,
2008b; Cannavo et al., 2013; Tully et al., 2012), which have high inherent nutrient contents,
highly porous soils and high infiltration rates. The N leaching losses from our plantation were
also lower than in banana plantations, characterized by very high fertilization rates (Wakelin et
al., 2011; Armour et al., 2013).



The nutrients leached at 1.5 m depth should be considered lost from uptake of oil palm
roots, as majority of the root mass and the highest root density are in the top 0.5 m depth (Nelson
et al., 2006; Schroth et al., 2000; Kurniawan et al., 2018). The high leaching fluxes of $NO_3^-$ and
Al implied a risk of groundwater pollution. During the high drainage fluxes following
fertilization, $NO_3^-$ concentrations in soil-pore water reached to 20–40 mg $L^{-1}$ in the inter-row
(covering 67 % of the plantation area), which was close to the 50 mg $L^{-1}$ limit for drinking water
(WHO, 2011), and Al concentrations in soil-pore water exceeded the limit of 0.2 mg $L^{-1}$ in 60
% of the samples. Nevertheless, before reaching to streams and rivers, these $NO_3^-$ and Al
concentrations can be diluted by surface flow and retained in the soil along flow paths: $NO_3^-$
can be temporarily adsorbed in the deeper layers of highly weathered soils by its inherently
high anion exchange capacity (Harmand et al., 2010; Jankowski et al., 2018) and can be
consumed by denitrification (Wakelin et al., 2011). Riparian buffers can mitigate the transport
of these agricultural pollutants to streams (Luke et al., 2017; Chellaiah and Yule, 2018).
Restoring riparian buffers in former forests converted to oil palm plantations have been listed
as one sustainability criteria, endorsed by the Roundtable for Sustainable Palm Oil association
(RSPO, 2018), and may provide additional regulation services (Woodham et al., 2019).
**5 Conclusions**
Our findings show that nutrient leaching losses in an oil palm plantation differed among
management zones, as a result of fertilization, liming, mulching and of different drainage
fluxes. The reduction of management intensity, i.e. mechanical weeding with reduced
fertilization rates, was effective in reducing nutrient leaching losses without reduction in yield
at least during the first two years of this experiment (Darras et al., 2019). Long-term
investigation of this management experiment is important to get a reliable response of yield and
a holistic economic analysis, including valuation of regulation services. Greenhouse gas
emissions should also be quantified, as another important parameter of environmental footprint



of oil palm production. Our findings and these further investigations should be incorporated
into science-based policy recommendations such as those endorsed by the RSPO.



**Data availability**

All data of this study are deposited at the EFForTS-IS data repository (https://efforts-is.uni-
goettingen.de), an internal data-exchange platform, which is accessible to all members of the
Collaborative Research Center (CRC) 990. Based on the data sharing agreement within the
CRC 990, these data are currently not publicly accessible but will be made available through a
written request to the senior author.

**Author contribution**

GF performed the experiments, analysed the data and wrote the manuscript in consultation
with MDC. EV and MDC conceived and planned the experiment. XD helped carry out the
water model simulations. AT aided in field activities organization and granting collaborations
agreements. All authors contributed to the final version of the manuscript.

**Competing interests**

No conflict of interest to declare

**Aknowledgments**

This study was part of the project A05 in the CRC990-EFForTS, funded by the German
Research Foundation (DFG, Project ID: 192626868 – SFB 990). We acknowledge the
collaborations with PTPN VI plantation, Kevin Darras, and project Z01, for the
implementation and maintenance of this field experiment. We thank Christian Stiegler, with
project A03, for the climate data, and Eckart Priesack for the Expert-N water sub-model. We
especially thank our field and laboratory assistants for their valuable dedications in all field
and laboratory activities. This research was conducted under the research permit of Ministry
of Research and Technology of Indonesia, 539351/SIP/FRP/E5/Dit.KI/X/2016.



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



**Tables and figures**
**Table 1** Soil physical and biochemical characteristics (mean ± standard errors, $n = 4$ plots) in
the top 50 cm depth for each management zone, averaged across experimental treatments.
Means within a row followed by different letters indicate significant differences among
management zones (one-way ANOVA with Tukey HSD or Kruskal–Wallis H test with multiple
comparisons extension at $P \leq 0.05$). Bulk density measured in the top 10 cm of soil, whereas
all the other parameters are for the 0–50 cm soil depth: element stocks are the sum of the
sampled soil depths (0–5 cm, 5–10 cm, 10–30 cm and 30–50 cm) and the rest are depth-
weighted averages, calculated for each replicate plot. ECEC, effective cation exchange capacity

| Soil properties | | Palm circle | Frond-stacked area | Inter-row |
|---|---|---|---|---|
| Bulk density | g cm$^{-3}$ | 1.37 ± 0.01$^a$ | 0.89 ± 0.01$^b$ | 1.36 ± 0.01$^b$ |
| Soil organic C | kg m$^{-2}$ | 6.2 ± 0.6$^b$ | 9.1 ± 0.8$^a$ | 6.4 ± 0.2$^b$ |
| Total N | g m$^{-2}$ | 402 ± 31$^b$ | 571 ± 39$^a$ | 426 ± 15$^{ab}$ |
| soil C:N ratio | | 15.5 ± 0.5$^a$ | 15.7 ± 0.3$^a$ | 15.0 ± 0.5$^a$ |
| $^{15}$N natural abundance | ‰ | 5.9 ± 0.1$^a$ | 5.3 ± 0.2$^a$ | 5.7 ± 0.2$^a$ |
| pH | 1:4 (H$_2$O) | 5.05 ± 0.08$^a$ | 5.00 ± 0.08$^{ab}$ | 4.81 ± 0.05$^b$ |
| ECEC | mmol$_c$ kg$^{-1}$ | 35 ± 2$^a$ | 28 ± 2$^a$ | 18 ± 1$^b$ |
| Base saturation | % | 48 ± 3$^a$ | 46 ± 4$^a$ | 20 ± 2$^b$ |
| Aluminum saturation | % | 52 ± 4$^b$ | 50 ± 2$^b$ | 78 ± 2$^a$ |
| Mg | g m$^{-2}$ | 32 ± 3$^a$ | 28 ± 6$^a$ | 9 ± 1$^b$ |
| Ca | g m$^{-2}$ | 169 ± 21$^a$ | 157 ± 15$^a$ | 37 ± 5$^b$ |
| K | g m$^{-2}$ | 39 ± 13$^a$ | 13 ± 1$^b$ | 6 ± 1$^b$ |
| Na | g m$^{-2}$ | 1.5 ± 0.4$^a$ | 0.7 ± 0.2$^a$ | 0.6 ± 0.2$^a$ |
| Al | g m$^{-2}$ | 66 ± 4$^b$ | 71 ± 4$^{ab}$ | 87 ± 3$^a$ |





| | | | | |
|---|---|---|---|---|
| Fe | g m$^{-2}$ | 1.4 ± 0.2[a] | 1.8 ± 0.4[a] | 1.8 ± 0.5[a] |
| Mn | g m$^{-2}$ | 0.7 ± 0.1[b] | 1.8 ± 0.3[a] | 0.6 ± 0.2[b] |
| H | g m$^{-2}$ | 0.2 ± 0.0[a] | 0.2 ± 0.0[a] | 0.2 ± 0.1[a] |


**Table 2** Annual water balance simulated from March 2017 to February 2018 for each
management zone.

| Water flux (mm yr$^{-1}$) | Palm circle | Frond-stacked area | Inter-row |
|---|---|---|---|
| Precipitation | 2772 | 2772 | 2772 |
| Transpiration | 828 | 448 | 401 |
| Evaporation | 228 | 214 | 434 |
| Interception | 351 | 209 | 209 |
| Runoff | 338 | 0 | 216 |
| Drainage (at 1.5 m depth) | 556 | 1806 | 1179 |






**Table 3** Annual leaching losses at 1.5 m depth for each experimental treatment from March
2017 to February 2018. Values are area-weighted averages of leaching losses in each
management zone (mean ± standard error, $n = 4$ plots). Means followed by different letters
indicate differences among experimental treatments (linear-mixed effect models on monthly
values followed by Tukey HSD test for multiple comparisons at $P \leq 0.05$). Treatments: ch =
conventional fertilization–herbicide; cw = conventional fertilization–mechanical weeding; rh =
reduced fertilization–herbicide; rw = reduced fertilization–mechanical weeding. DON =
dissolved organic N; TDN = total dissolved N.

| Element leaching (kg ha$^{-1}$ yr$^{-1}$) | ch | cw | rh | rw |
|---|---|---|---|---|
| $NO_3^-$-N | 71.5 ± 20.1[a] | 48.2 ± 13.0[ab] | 36.3 ± 20.1[b] | 30.0 ± 5.7[b] |
| $NH_4^+$-N | 1.7 ± 0.2[a] | 1.7 ± 0.1[a] | 1.8 ± 0.1[a] | 1.7 ± 0.2[a] |
| DON | 0.5 ± 0.5[a] | 0.6 ± 0.3[a] | 0.4 ± 0.1[a] | 0.3 ± 0.0[a] |
| TDN | 73.6 ± 20.2[a] | 50.4 ± 13.1[ab] | 38.4 ± 8.9[b] | 32.0 ± 5.8[b] |
| Ca | 26.6 ± 4.3[a] | 19.4 ± 4.4[b] | 18.2 ± 1.8[b] | 17.0 ± 2.1[b] |
| Mg | 11.6 ± 2.5[a] | 7.7 ± 0.8[b] | 9.1 ± 0.7[ab] | 10.8 ± 3.6[ab] |
| K | 8.1 ± 1.3[a] | 6.2 ± 0.7[b] | 8.9 ± 0.6[a] | 5.7 ± 1.1[b] |
| Na | 15.9 ± 3.5[ab] | 13.6 ± 2.4[b] | 18.9 ± 3.1[a] | 13.1 ± 1.2[b] |
| Mn | 0.3 ± 0.1[a] | 0.2 ± 0.0[b] | 0.2 ± 0.0[bc] | 0.1 ± 0.0[c] |
| Total Al | 40.8 ± 11.5[a] | 20.8 ± 7.6[b] | 19.9 ± 6.8[b] | 21.8 ± 3.1[b] |
| Total S | 2.4 ± 0.5[a] | 1.8 ± 0.4[a] | 2.1 ± 0.6[a] | 4.9 ± 3.3[a] |
| Total Fe | 0.2 ± 0.0[a] | 0.5 ± 0.3[a] | 0.2 ± 0.0[a] | 0.5 ± 0.3[a] |
| Total P | 0.0 ± 0.0[a] | 0.1 ± 0.0[a] | 0.0 ± 0.0[a] | 0.0 ± 0.0[a] |




| | | | | |
|---|---|---|---|---|
| Cl | 79.7± 15.8[a] | 36.9 ± 8.3[b] | 67.7 ± 8.7[a] | 78.3 ± 7.5[a] |



**Table 4** N and base cation retention efficiencies in the soil for each management zone and
experimental treatment (means ± standard error, $n$ = 4 plots). Means followed by different
lowercase letters indicate differences among experimental treatments for each management
zone, whereas different uppercase letters indicate differences among management zones for
each experimental treatment (one-way ANOVA with Tukey HSD or Kruskal–Wallis H test
with multiple comparisons extension at $P \leq 0.05$). Weighted-average is based on the areal
coverage of each management zone: 18 % for palm circle, 15 % for frond-stacked area, and 67
% for inter-row. Treatments: ch = conventional fertilization–herbicide; cw = conventional
fertilization–mechanical weeding; rh = reduced fertilization–herbicide; rw = reduced
fertilization–mechanical weeding. See section 2.4 for calculations of N and base cation
retention efficiency.

| | ch | cw | rh | rw |
|---|---|---|---|---|
| **N retention efficiency (mg N m$^{-2}$ d$^{-1}$ / mg N m$^{-2}$ d$^{-1}$)** | | | | |
| Palm circle | 0.987 ± 0.002[a A] | 0.982 ± 0.007[a AB] | 0.986 ± 0.003[a AB] | 0.997 ± 0.000[a A] |
| Frond-stacked area | 0.984 ± 0.004[a A] | 0.989 ± 0.004[a A] | 0.993 ± 0.001[a A] | 0.987 ± 0.002[a A] |
| Inter-row | 0.877 ± 0.025[a B] | 0.870 ± 0.022[a B] | 0.900 ± 0.018[a B] | 0.906 ± 0.039[a A] |
| Weighted-average | 0.925 ± 0.022[a] | 0.934 ± 0.020[a] | 0.945 ± 0.012[a] | 0.946 ± 0.018[a] |
| **Base cation retention efficiency (mol$_c$ m$^{-2}$ yr$^{-1}$ / mol$_c$ m$^{-2}$ yr$^{-1}$)** | | | | |





| | | | | |
|---|---|---|---|---|
| Palm circle | $0.967 \pm 0.008^{ab\ A}$ | $0.982 \pm 0.002^{a\ A}$ | $0.937 \pm 0.013^{b\ A}$ | $0.974 \pm 0.010^{ab\ A}$ |
| Frond-stacked area | $0.884 \pm 0.013^{b\ A}$ | $0.950 \pm 0.004^{a\ A}$ | $0.960 \pm 0.002^{a\ A}$ | $0.928 \pm 0.016^{ab\ A}$ |
| Inter-row | $0.588 \pm 0.086^{b\ B}$ | $0.875 \pm 0.022^{a\ B}$ | $0.704 \pm 0.048^{ab\ B}$ | $0.822 \pm 0.063^{ab\ A}$ |
| Weighted-average | $0.876 \pm 0.009^{b}$ | $0.945 \pm 0.007^{a}$ | $0.902 \pm 0.019^{ab}$ | $0.934 \pm 0.012^{ab}$ |






**Figure 1** Pearson correlation test between modeled (red line) and field-measured soil water
matric potential (black points) ($n$ = 50 field measurements over one year) for each management
zone at 30 and 60 cm depths.

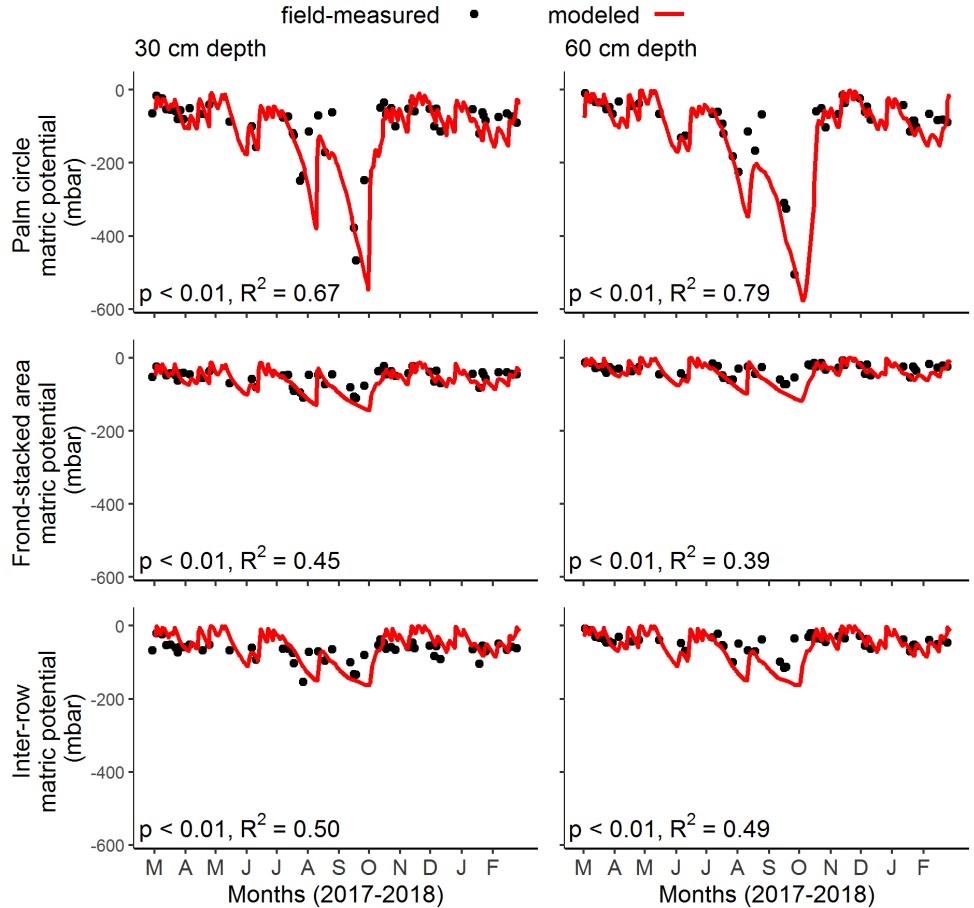






**Figure 2** Monthly water drainage at 1.5 m depth, simulated in each management zone, and
daily rainfall from March 2017 to February 2018. The gray shaded area represent the dry season
(precipitation < 140 mm month$^{-1}$)

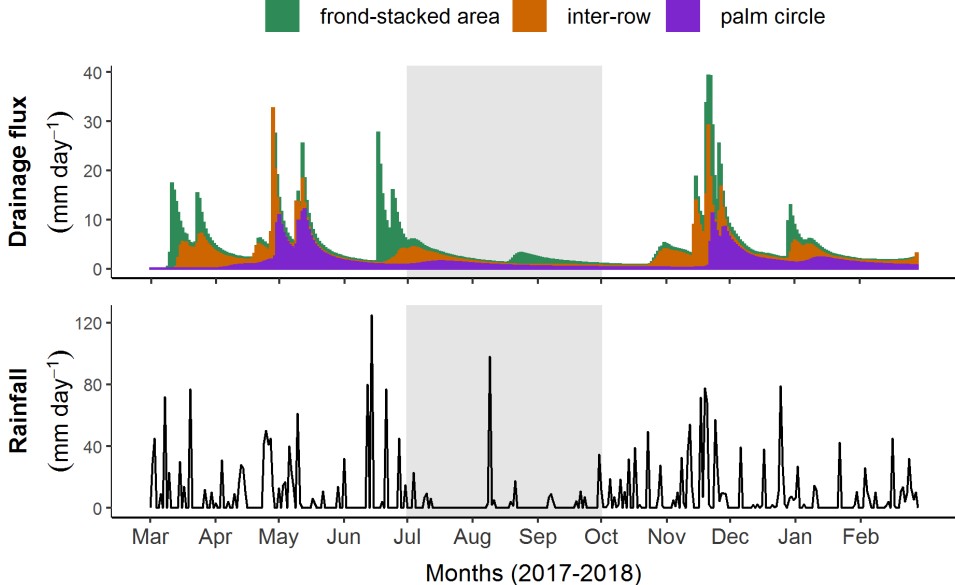




**Figure 3.** Partial cation-anion charge balance of the major solutes (with concentrations > 0.03 mg L$^{-1}$) in soil water at 1.5 m depth for each experimental treatment in the different management zones. The concentrations of organic acids (RCOO$^-$) and carbonates (HCO$_3^-$) are calculated as the difference between the measured cations and anions. Treatments: ch = conventional fertilization–herbicide; cw = conventional fertilization–mechanical weeding; rh = reduced fertilization–herbicide; rw = reduced fertilization–mechanical weeding.

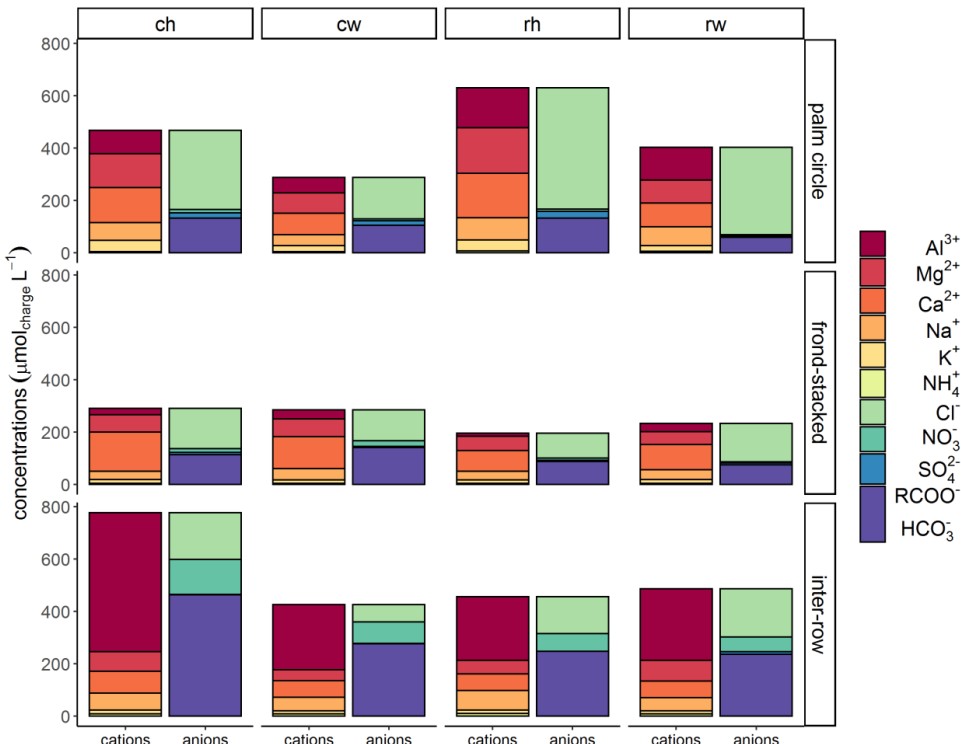



**Figure 4** Monthly leaching losses at 1.5 m depth (mean ± standard errors, $n$ = 4 plots) for each management zone. Black arrows indicate fertilizer applications and the gray shaded area represents the dry season (precipitation < 140 mm month$^{-1}$).

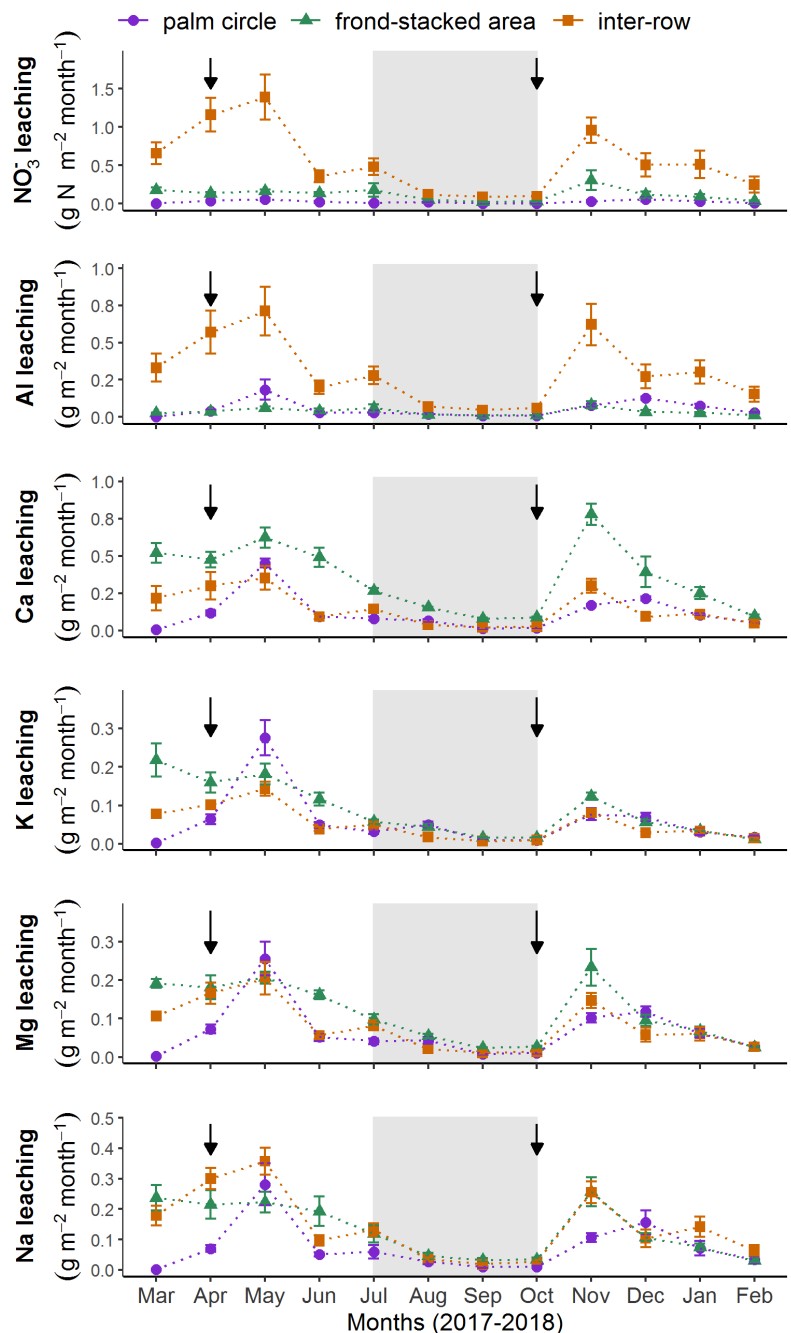

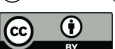



**Figure 5** Average monthly leaching losses at 1.5 m depth for each experimental treatment from
March 2017 to February 2018. Values are area-weighted averages of leaching losses in each
management zone (means ± standard errors, *n* = 4 plots). For each parameter, different letters
indicate significant differences among treatments (linear-mixed effect models on monthly
values followed by Tukey HSD test for multiple comparisons at $P \le 0.05$). Treatments: ch =
conventional fertilization–herbicide; cw = conventional fertilization–mechanical weeding; rh =
reduced fertilization–herbicide; rw = reduced fertilization–mechanical weeding

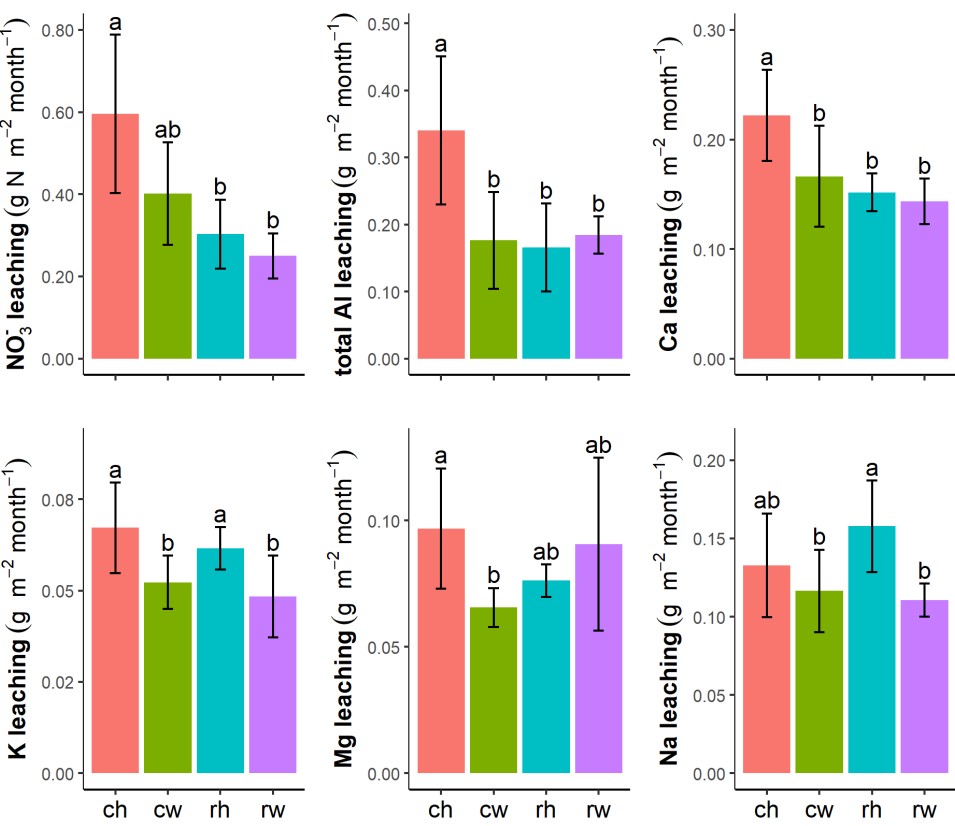






**Appendices**
**Table A1** Gross N mineralization rates (means ± SE, $n = 4$ plots) in the top 5 cm soil for each
treatment and management zone in a large-scale plantation in Jambi, Indonesia. Measurements
were done on intact soil cores in February 2018 using the $^{15}N$ pool dilution technique, as
described in details by Allen et al. (2015). Treatments: ch = conventional fertilization–
herbicide; cw = conventional fertilization–mechanical weeding; rh = reduced fertilization–
herbicide; rw = reduced fertilization–mechanical weeding

| Gross N mineralization (mg N m$^{-2}$ d$^{-1}$ ) | | | | |
|---|---|---|---|---|
| | ch | cw | rh | rw |
| palm circle | 2.2 ± 0.6 | 1.9 ± 0.4 | 1.8 ± 0.6 | 3.4 ± 0.2 |
| frond-stacked area | 22.4 ± 3.3 | 32.5 ± 8.0 | 22.4 ± 7.2 | 16.6 ± 5.2 |
| inter-row | 4.8 ± 1.1 | 4.0 ± 0.6 | 3.8 ± 0.8 | 4.4 ± 0.9 |

*Note:* These data are not included in the manuscript to avoid double-publication as these
results were reported in our previous study (Formaglio et al., unpublished data).



**Table A2** Literature comparison of annual N fertilization and total N leaching losses across
tropical plantations.

| Author | Soil type | rainfall (mm yr$^{-1}$) | Type of plantation management | N applied (kg ha$^{-1}$ yr$^{-1}$) | Total N leaching (kg ha$^{-1}$ yr$^{-1}$) | Percentage N leached (%) |
|---|---|---|---|---|---|---|
| Present study | loam Acrisol | 2772 | intensive oil palm | 260 | 74 | 28 |
| Present study | loam Acrisol | 2772 | intensive oil palm | 130 | 38 | 28 |
| Omoti et al. 1983 | sandy clay Acrisol | 2000 | intensive oil palm | 150 | 9 | 6 |
| Kurniawan et al. 2018 | loam Acrisol | 3418 | smallholder oil palm | 88 | 11 | 12.5 |
| Tung et al. 2009 | Acrisol | - | intensive oil palm | 128 | 3 (150 days) | 2 |
| Tung et al. 2009 | Acrisol | - | intensive oil palm | 251 | 3 (150 days) | 1 |
| Banabas et al. 2008 | clay loam Andosol | 2398 | intensive oil palm | 100 | 37 | 37 |
| Banabas et al. 2008 | sandy loam Andosol | 3657 | intensive oil palm | 100 | 103 | 103 |
| Cannavo et al. 2013 | clay loam Andosol | 2678 | coffee agroforestry | 250 | 157 | 63 |



| Tully et al., 2012 | clay loam Andosol | 2700 | coffee agroforestry | 120 | 119 | 99 |
| Armour et al. 2013 | clay Acrisol | 1958 | intensive banana | 476 | 164 | 34 |
| Wakelin et al. 2011 | loam Acrisol | 2685 | intensive banana | 305 | 116 | 38 |



**Figure A1** Lysimeter locations at each treatment plot, with two subplots (blue rectangles) that
each included the three management zones (blue crosses): 1) lysimeters in the palm circle
were at 1 m from the palm trunk, 2) in the frond-stacked area, at about 4 m from the palm
trunk, and 3) in the inter-row, at approximately 4 m from the palm trunk.

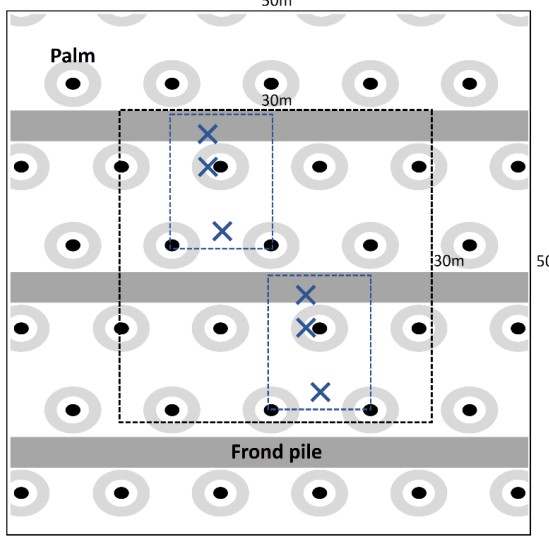
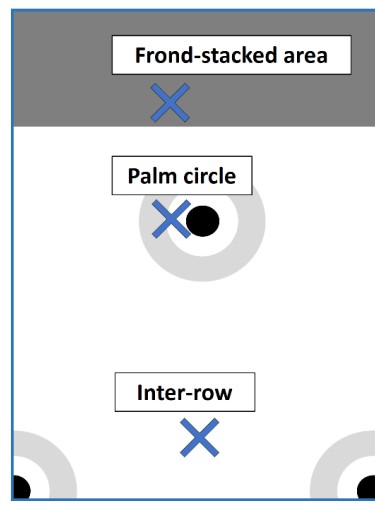

