# Peer review of "Herbicide weed control increases nutrient leaching compared to mechanical"

_Biogeosciences, 2020_

## Referee Comment (RC1) · Anonymous Referee #1 · 16 Jul 2020

Thank you for submitting an interesting manuscript on nutrient leaching in oil palm plantations under different weeding and fertiliser regimes. Your results are valuable for providing evidence for assessing more environmentally friendly options of growing oil palm. I support publication of this manuscript after revision.

My main comment is that the analysis/results of yield are not appropriately considered. The authors refer to previously published studies on the yield aspect but do not actually quote any data/numbers on yield. This is an important point that is missing. Any management options leading to more environmentally friendly or sustainable growing of oil palm will ultimately be scrutinised under the yield aspect. So it is important to include

the actual figures in this assessment (even if they have been published elsewhere). Just saying there was no difference is not sufficient. Growers would like to know the actual yield to see that you were not working on a plantation with unusually high or low yield anyway which might have masked any management effects. So a revised version needs results and discussion sections that are expanded with details on yield.

In addition I have a few general and specific technical comments/corrections.

General: There should be no space between number and %. Please revise this throughout the manuscript. The term 'stem' might be more appropriate to use than 'trunk'. Please replace 'trunk' with 'stem' throughout the manuscript. The term 'conventional' is a bit misleading. Perhaps 'standard practice' or 'standard industry practice' would be a more appropriate term to use? Generally spelling and grammar need to be checked carefully, some sentences are too long and convoluted.

Specific: Title remove 'as' I 39 replace 'have' with 'has' I 40 remove 'and' I 45 remove space between 57 and % (see general comment above) I 48 introduce N as nitrogen I 59 introduce P as phosphor NO3 as nitrate I 61 comma after reference before 'whereas' I 68 remove 'of' in front of oil palm I 80 herbicides needs to be plural I 82 'herbicide weeding' perhaps clarify that this is chemical weeding with herbicides as supposed to mechanical weeding I 97 circles needs to be plural I 101 canopy interception will depend on the age of the plantation and whether there is canopy closure or not. please elaborate here and say that there will be a difference between younger and older plantations I 131 Was the plantation terraced? This should be a discussion point generally as nutrient flows will be different in plantations with even terrain as supposed to terraced plantations. It seems to become more popular (also in Indonesia) to terrace plantations (when replanting) even if the terrain is not that hilly to begin with. This might have potentially large implications on nutrient leaching. I 135 remove space between number and % I 137 chemically or mechanically weeded? Please clarify I 142 Is the fertiliser applied in pellets or granules? Broad spread? Please give details I 145 Were no EFB (empty fruit bunches) returned to the plantation? This is common practice and

would add more organic matter to the plantation in addition to the palm fronds. if it wasn't done in your plantation, it still needs to be discussed in the discussion section. l 170 replace 'till' with 'to' l 182 check reference, should only be (2018) in brackets l 191 not clear what (2018) is suppose to mean? Should there be a reference? And do you mean combined bicarbonate and organic acids? Please clarify (also in Fig 3) l 205 do you have anything to base your assumption on? Any measurements/references? l 209 insert 'The' in front of Expert-N model l 211 insert 'the' after using l 217 add reference for Richards' equation l 224 'stem' instead of 'trunk' l 237 space between 1 and um l 246 remove 'to' in front of 192 l 306 replace 'high' with 'strong' l 318 specify here the months for typical dry and wet season l 424 But as you are saying elsewhere, they would not fertilise in the dry season either. So you might have to elaborate here. l 438 Mention EFB, POME, compost or a combination of these as they are all commonly used types of organic fertiliser in oil palm plantations l 443 circle closest to the roots for direct uptake of the fertilisers? l 453 replace 'highest' with 'higher' l 458 trunk = stem l 464 insert 'will' in front of largely l 492 replace 'increased' with 'increases' l 494 The last part of the sentence doesn't fit the first. Please rephrase. l 502 replace 'influence' with 'effect' l 505 Please rephrase this sentence, it sounds a bit clumsy and i is not that clear what you are trying to say. l 508 Replace 'have' with 'has' l 519 Start a new sentence after the reference as the second half doesn't follow the first. l 525 replace 'conventional' with 'standard practice'? l 527 This is where you need to expand on the yield aspect and include data etc. l 553 Insert 'the' in front of majority l 559 Remove 'to' in from of streams

Table 1 call this soil physicochemical parameters as it's not really biochemical Table 4 Did you quote the decimals to significant figures? Figures 2,3 and 5 either call the x axis 'month' or remove 'months' all together as it is obvious that is is a data axis (just add years) Figure 3 are RCOO and HCO3 combined or separate? Please clarify. If it is combing perhaps say '+'? l 1010 with 'unpublished' do you mean not yet published? If it is in prep, please add, otherwise remove this citation. You are contracting 'reported' with 'unpublished'.

[Figure]

---

## Referee Comment (RC2) · Anonymous Referee #2 · 18 Jul 2020

General comments This manuscript provides important results for the management of oil palm plantations, showing that mechanical weeding reduces leaching losses compared to conventional practices using herbicides. This study is one of the first to take into account the great spatial heterogeneity of oil palm plantations to estimate nutrient leaching. Although the experimental design of the factorial management experiment is strong with big plots and 4 blocks, there are some weaknesses in the methodology: only one year of soil solution sampling, only one depth of soil solution collection, water transfer model parameterized using data in the literature (LAI, fine root densities, soil hydraulic parameters). However, studies that accurately measure nutrient leaching over several years are rare in the tropics and the data set presented here is original

and timely. Moreover, the discussion section shows that the magnitude of the drainage fluxes is consistent with evapotranspiration rates estimated in oil palm plantations from various methodologies. The manuscript is generally well written. Only minor revisions are recommended prior to publication.

Specific comments

Abstract: It would be interesting to indicate some values of nutrient losses in plots with conventional management practices. L32: "Our findings signified that mechanical weeding..." should be replaced by "Our findings suggested that mechanical weeding..." because you cannot generalize to Indonesia your results at a single site.

Introduction: The Introduction section is clear and informative. L71-73: Not always the case, see for example eucalypt plantations.

Material and methods The lysimetry design was suitable for the quantification of leaching losses. You might indicate that, even though the time period for soil stabilization was short in your study (only two months from the installation of the ceramic cups to the start of the soil solution collection and four months from the implementation of the factorial management experiment to the collection of the soil solutions), this period was sufficient because the biological processes are rapid in tropical soils.

You may be interested in two articles that accurately describe the spatial development of roots in oil palm plantations: Plant and Soil 189: 33-48, 1997 and Plant and Soil 190: 235-246, 1997.

More information should be given on the water drainage model. How have you dealt with run off from one management zone to another? The parameterization of the model is rough, without measurements of root profiles and soil hydraulic parameters in each treatment and management zones. Moreover, the validation from soil water potentials is also rough with only two depths (it would have been interesting to include the depth of 1.5 m where soil solutions are collected) and punctual measurements in only

two treatments (only 12 tensiometers). It is important to provide a table (in appendix) showing the values given to all the parameters used in the Expert-N water sub-model for each management zone.

Statistical analyses are clearly presented.

Results No specific recommendation, this section is well written.

Discussion You might be interested by a recent paper providing values for N leaching in oil palm plantations. The three management zones were sampled in the field to validate this model and the order of magnitude is consistent with your results: DOI: 10.1002/agj2.20109 The comparison with other perennial tropical plantations is interesting (Table A2). Could you add forest plantations to this appendix (pines, eucalypts, acacias) and rubber plantations if you find data in the literature?

L515-517: not sure that the amounts of Na taken up by soil macrofauna could be sufficient to explain the differences. It has been demonstrated that oil palm can take up Na in addition/substitution to K (Bonneau, X., Boutin, D., Bourgoing, R., Sugarianto, J., 1997. Le chlorure de sodium, fertilisant idéal du cocotier en Indonésie. Plantations, Recherche, Développement 4 (5), 336–346), as also shown recently in eucalypt plantations.

Tables, Figures and appendices The 4 tables, 5 figures and 3 appendices shown are clear and relevant.

Technical corrections

L329: than in the inter-row L332: dissolved organic N or total dissolved N? L453: higher?
* * *

---

## Referee Comment (RC3) · Anonymous Referee #3 · 23 Jul 2020

Peer review of, "Herbicide weed control increases nutrient leaching as compared to mechanical weeding in a large-scale oil palm plantation", which was submitted for a possible publication to Biogeosciences (bg-2020-153).

General comments The manuscript deals with the measurements of nutrient leaching in an oil palm plantation, trying to clarify the effects of fertilization rate and weeding methods on various nutrients leaching. The authors have collected data between 2017-2018. Main findings of the authors are described below. The reduced management intensity (e.g., mechanical weeding and less fertilization rates) could effectively reduce nutrient leaching losses. In addition, the nutrient leaching is significantly different in

different management zones, such as inter-row, palm circle and frond-stacked area, in oil palm plantations. These results are very important to researchers and future study, even to the policy makers. I highly appreciate the presented research in this text. The paper is well organized, figures and tables are carefully prepared. The large amount of the field work and interesting results performed by the authors are worthy of publication. However, I found there are a few questions in the manuscript after reading thoroughly. I have several specific comments that should be adequately incorporated and explained by the authors before this manuscript is considered for publication. These comments are detailed below.

Specific comments The abstract should be revised. 1. Line 23-28 Could you consider the specific data (e.g., low solute concentrations, small drainage…) should be added, and thus increasing the persuasiveness for the readers? (Maybe it is important in the Abstract)

The introduction is very good. The scientific question is clear. 1. Line 41 The term "e.g" should be "e.g." 2. Line 52-54 Could you provide some references? Thank you. 3. Line 57-63 Indeed, high precipitation rate is a critical driver for surface runoff and associated nutrient losses. Particularly for considerable plantations. However, the leaching losses may be offset by a high nutrient cycling due to the rapid uptake of plants. 4. Line 92 How far the radius of the palm circle? 5. Line 98-99 Could you describe more details about the root distribution of oil palm? I am not sure the roots of palm only grow around the trunk. I think the root biomass between inter-row area may be high in somewhere.

The Materials and methods section is good structure. The content is detailed and makes it easy for readers to understand. 1. Line 159 Replace the "x" between "the 50 x 50 m" with "×". 2. Line 160 Where is the plant materials from the mechanical weeding? Are they transported far away from the plots? 3. Line 232 Is the runoff set to 0? Do you mean "no overland runoff"? 4. Line 243 Soil physical-chemical characteristics. 5. Line 247 See comment 7. 6. Line 265 Please simplify the statistical analyses section.

The results section is well-organized manner. However, some statements are so long that they (e.g., the section "3.2 Differences in leaching losses...") should be simplified to delete some non-key contents. 1. Line 310-311 The drainage flux is low. Do you investigate the stem flow (may be influenced by "funnel effect" of canopy of oil palm?)? Some studies demonstrated that the infiltration was enhanced around the tree trunk. 2. Line 346 Why is different between the various elements leaching?

The discussion section was carefully written and prepared. 1. Line 391 I recommend the ratios of runoff/interception/evaporation/transpiration to precipitation was supplemented in the Table 2 for better understanding. 2. Line 434-438 How to understand the use of organic amendments and slow-release fertilizers? E.g., mulching application? Under the high temperature and precipitation in some tropical areas, the plant materials decompose quickly, and the litterfall may have very short residence time on the ground. Could you provide any information on the standing plant litter in your treatments? Thank you! 3. Line 552-553 Although the mechanical weeding is sustainable way in ecological view, the farmers were reluctant to adopt due to its money-consuming and labour-consuming. Undoubtedly, mechanical weeding is a promising measure to reduce nutrient leaching.

---

## Author Comment (AC1) · 10 Aug 2020

RC> My main comment is that the analysis/results of yield are not appropriately considered. The authors refer to previously published studies on the yield aspect but do not actually quote any data/numbers on yield. This is an important point that is missing. Any management options leading to more environmentally friendly or sustainable growing of oil palm will ultimately be scrutinised under the yield aspect. So it is important to include the actual figures in this assessment (even if they have been published elsewhere). Just saying there was no difference is not sufficient. Growers would like to know the actual yield to see that you were not working on a plantation with unusually

high or low yield anyway which might have masked any management effects. So a revised version needs results and discussion sections that are expanded with details on yield.

AR> Thank you for this observation. We agreed that the focus on the yield is essential for the development of long-term more sustainable management practices. We will include appendix Fig A2 with the yield measured up to three years after the establishment of the experiment. We will expand the discussion about the yield in l 518 as follows: "The yield, measured up to three years after the establishment of the experiment, was on average 30 Mg of fresh fruit bunches ha-1 yr-1 and it was comparable among experimental treatments (Figure A2, Darras et al. 2019). This indicates that the reduced management intensity did not affect productivity in the first three years, but the long-term monitor is essential as it may take a longer time for the yield to respond to the treatments (Tao et al. 2017). Also, the cost of the two weeding treatments (i.e. herbicide vs mechanical) was comparable because it is standard practice to combine the use of herbicide with the periodic mechanical cutting of resistant ground vegetation (Darras et al., 2019; Pahan, 2010). Therefore, these results altogether advocate for the higher sustainability of mechanical weeding over herbicide application".

[Figure A2 Annual yield for each experimental treatment from 2017 to 2019. Treatments: ch = conventional fertilization–herbicide; cw = conventional fertilization–mechanical weeding; rh = reduced fertilization–herbicide; rw = reduced fertilization–mechanical weeding. Notes: yield was measured by weighting the harvested fresh fruit bunches from each palm in the inner 30 x 30 m area of each plot.]

RC>There should be no space between number and %. Please revise this throughout the manuscript.

AR> We will it correct in the revised manuscript.

RC> The term 'stem' might be more appropriate to use than 'trunk'. Please replace 'trunk' with 'stem' throughout the manuscript.

AR> We will replace it in the revised manuscript.

RC> The term 'conventional' is a bit misleading. Perhaps 'standard practice' or 'standard industry practice' would be a more appropriate term to use?

AR>We prefer to keep the term of conventional practice for consistency with other manuscripts published on this experiment, i.e. Darras et al. 2019, and others in preparation.

RC> Generally spelling and grammar need to be checked carefully, some sentences are too long and convoluted.

AR> Based on this comment we revised the manuscript and decide to modify the structure of a few sentences to improve clarity.

L 67 "Oil palm plantations can possibly have low leaching losses, as a consequence of high evapotranspiration and thus low drainage fluxes (Tarigan et al., 2020). However, most oil palm plantations are large-scale enterprises that are characterized by intensive management with high fertilization rates and herbicide application".

L 343 "On the other hand, base cation leaching fluxes had opposite patterns as their concentrations as Ca, K, and Mg leaching were higher in the frond-stacked area than the palm circle and inter-row (all P < 0.01; Fig. 4)".

L 393 "Considering the areal proportions of the three management zones, the weighted-average drainage flux (1161 mm yr-1) was lower than that estimated for smallholder oil palm plantations near our study site (1614 mm drainage 396 flux with 3418 mm precipitation measured in 2013; Kurniawan et al., 2018). However, their ratios to annual precipitation were comparable, due to the relatively wet year of 2013. Also, evapotranspiration rate is higher in large-scale than smallholder oil palm plantations in our study area (Röll et al., 2019), which would lead to lower drainage flux in large-scale plantation".

L 410 "High drainage might have stimulated the downward transport of elements and

decreased their residence time in the soil, and thus their adsorption onto the soil exchange sites (Lohse and Matson, 2005). At the same time, high water fluxes usually dilute the element concentrations in the soil-pore water; high concentrations were maintained because of fertilizer and lime applications in the same periods, resulting in parallel peaks of drainage and leaching fluxes (Figs. 2 and 4)".

L 441 "Contrary to our first hypothesis, leaching losses among management zones were generally higher in the inter-row, especially for mineral N (largely NO3; Fig. 3), and lower in the palm circle (Fig. 4)".

L 460 "The main accompanying cation for NO3- leaching in the inter-rows was Al3+ (Figs. 3 and 4). This is because this zone's soil pH (Table 1) was within the Al-buffering range (pH 3–5; van Breemen et al., 1983) due to no direct lime application and thus low base saturation (Table 1)".

L 489 "Thus, the larger base cations leaching in the frond-stacked area compared to the inter-row (Fig. 4) merely mirrored their high exchangeable levels (Table 1)".

RC> Title remove 'as'.

AR> We will change this.

RC> l 39 replace 'have' with 'has'.

AR> We will correct this in the revised manuscript.

RC> l 40 remove 'and' .

AR> We will correct this in the revised manuscript.

RC> l 45 remove space between 57 and % (see general comment above).

AR> We will correct this in the revised manuscript.

RC> l 48 introduce N as nitrogen, l 59 introduce P as phosphor NO3 as nitrate.

AR> We will insert "phosphor" and "nitrate" in the revised manuscript.

RC> l 61 comma after reference before 'whereas'.

AR> We will correct this in the revised manuscript.

RC> l 68 remove 'of' in front of oil palm.

AR> We will correct this in the revised manuscript.

RC>l 80 herbicides needs to be plural.

AR> We will correct this in the revised manuscript.

RC> l 82 'herbicide weeding' perhaps clarify that this is chemical weeding with herbicides as supposed to mechanical weeding.

AR> The sentence will be modified in the manuscript as "Chemical weeding with herbicide is commonly practiced in large-scale oil palm plantations: herbicide is placed in the area where the fertilizers are applied, to reduce competition for nutrients and water with ground vegetation, and in the inter-rows, to facilitate access during harvest (Corley and Tinker, 2016)".

RC> l 97 circles needs to be plural.

AR> We prefer to keep it singular for consistency, as it is the management zone mentioned above (l 92).

RC> l 101 canopy interception will depend on the age of the plantation and whether there is canopy closure or not. Please elaborate here and say that there will be a difference between younger and older plantations.

AR> We agree that the age of the palm is important to determine the level of interception in the inter-row. However, we think that this sentence doesn't need to be modified because the interception in the inter-row will always be lower than in the palm circle, independently of the age of the palm. Even when the canopy closes, 7-8 years after planting, the interception is lower in the inter-row than in the palm circle (Banabas et

al. 2008).

RC> l 131 Was the plantation terraced? This should be a discussion point generally as nutrient flows will be different in plantations with even terrain as supposed to terraced plantations. It seems to become more popular (also in Indonesia) to terrace plantations (when replanting) even if the terrain is not that hilly to begin with. This might have potentially large implications on nutrient leaching.

AR> The plantation was not terraced and it was not on a slope position. We will add at l 133 "The plantation was not terraced".

RC> l 135 remove space between number and %.

AR> We will correct this in the revised manuscript.

RC> l 137 chemically or mechanically weeded? Please clarify.

AR> We will specify that they were chemically weeded, as it is the standard practice in the investigated plantation.

RC> l 142 Is the fertiliser applied in pellets or granules? Broad spread? Please give details.

AR> The fertilizer was in the form of granulates and it was broad spread within the palm circle. We will specify this in the updated version of the manuscript.

RC> l 145 Were no EFB (empty fruit bunches) returned to the plantation? This is common practice and would add more organic matter to the plantation in addition to the palm fronds. If it wasn't done in your plantation, it still needs to be discussed in the discussion section.

AR> In our studied plantation there is a certain amount of EFB returned to the plantation (mainly in the form of compost). The 2025-ha plantation investigated is owned by the company PTPN6, which has a total of 90122 ha of oil palm plantations in Jambi province. Once the fruits are processed in the mill, the EFB (also in the form of compost

or ashes) is redistributed across all the plantations of PTPN6 in a rotation. In addition, within the same plantation, the application of EFB follows a rotation system based on management blocks. Therefore the timing and the amount of EFB distributed are quite complicated to predict. In our experimental plots, we did not include the EFB compost in the treatments. This is for two reasons: 1) our experimental plots encompassed different management blocks and therefore the different timing of the EFB application might have biased the results; 2) the aim of the experiment is to compare the most standardized management practices with the reduced of management intensity, and the application of EFB is not done regularly and in all the plantations (it is normally applied just to the area next to the mill). Since we didn't include the practice of using EFB in our experiment, we cannot discuss its effects on leaching, but we will hint to it in l 438 as follows: "One other option is the use of organic amendments, such as empty fruit bunches, compost, palm oil mill effluent, or slow-release fertilizers, which have been shown to reduce N leaching in tropical cropping systems (Nyamangara et al., 2003; Mohanty et al., 2018; Steiner et al., 2008). In addition, organic fertilizer can improve soil fertility in oil palm plantations (Comte et al., 2013; Boafo et al.,2020), as was also evident with mulching of senesced oil palm fronds (i.e. high SOC, total N, ECEC and base saturation in the frond-stacked area; Table 1)".

RC> l 170 replace 'till' with 'to'.

AR> We will replace in the revised manuscript.

RC> l 182 check reference, should only be (2018) in brackets.

AR> Thank you for noticing this, it will be corrected in the revised manuscript.

RC>l 191 not clear what (2018) is suppose to mean? Should there be a reference? And do you mean combined bicarbonate and organic acids? Please clarify (also in Fig 3).

AR> Yes the (2018) was supposed to be the reference "Kurniawan et al. 2018", it will

be corrected in the revised manuscript. We will also add the word "combined" to avoid confusion.

RC> l 205 do you have anything to base your assumption on? Any measurements/references?

AR> For example, Khasanah et al., Agriculture, Ecosystems and Environment 211 (2015) 195–206 found no differences in bulk density among management zones. This reference will be included in the revised manuscript.

RC> l 209 insert 'The' in front of Expert-N model.

AR> We will insert it in the revised manuscript.

RC> l 211 insert 'the' after using.

AR> We will insert it in the revised manuscript.

RC> l 217 add reference for Richards' equation.

AR> We will add the reference in the revised manuscript.

RC> l 224 'stem' instead of 'trunk'.

AR> We will replace this in the revised manuscript.

RC> l 237 space between 1 and um.

AR> We will add a space in the revised manuscript.

RC> l 246 remove 'to' in front of 192.

AR> We will remove it in the revised manuscript.

RC> l 306 replace 'high' with 'strong'.

AR> We will replace it in the revised manuscript.

RC> l 318 specify here the months for typical dry and wet season.

AR> We think that specify the typical months will be confusing. The dry period is clearly indicated in the picture that is referred in the text.

RC> l 424 But as you are saying elsewhere, they would not fertilise in the dry season either. So you might have to elaborate here.

AR> It is true that the farmers don't fertilise in the dry season, which is also not advisable for the reasons explained in l 429 but they could fertilize for example one month later, avoiding the peaks of drainage. Unfortunately, as explained in the manuscript, this is not practical because it is difficult to predict the period of high drainage fluxes from the precipitation pattern and so the most viable solution to reduce leaching is to reduce fertilization rates and increase nutrient retention efficiency.

RC> l 438 Mention EFB, POME, compost or a combination of these as they are all commonly used types of organic fertiliser in oil palm plantations.

AR> This will be mentioned in the revised manuscript (see answer to comment to l 145).

RC> l 443 circle closest to the roots for direct uptake of the fertilisers?

AR> We prefer to explain this in detail later in the manuscript (starting at l 466).

RC> l 453 replace 'highest' with 'higher'.

AR> We will correct this in the revised manuscript.

RC> l 458 trunk = stem.

AR> We will replace it in the revised manuscript.

RC>l 464 insert 'will' in front of largely.

AR> We will insert it in the revised manuscript.

RC> l 492 replace 'increased' with 'increases'.

AR> We will correct this in the revised manuscript.

RC> l 494 The last part of the sentence doesn't fit the first. Please rephrase.

AR> We will remove the last part of the sentence for more clarity, as the study is not investigating soil fertility.

RC> l 502 replace 'influence' with 'effect'.

AR> We will replace this in the revised manuscript.

RC> l 505 Please rephrase this sentence, it sounds a bit clumsy and i is not that clear what you are trying to say.

AR> For more clarity, we will rephrase the sentence as follows: "In line with our second hypothesis, the weeding methods clearly influenced leaching losses: the mechanical weeding treatment had generally lower leaching fluxes than the herbicide treatment (Fig. 5; Table 5)."

RC> l 508 Replace 'have' with 'has'.

AR> We will correct this in the revised manuscript.

RC>l 519 Start a new sentence after the reference as the second half doesn't follow the first.

AR> We will modify this in the revised manuscript.

RC> l 525 replace 'conventional' with 'standard practice'?

AR> We prefer to keep the world "conventional" for consistency throughout the manuscript.

RC> l 527 This is where you need to expand on the yield aspect and include data etc.

AR> We decided that is was better to talk about the yield in l 518 where we first mention the economic aspects of the treatments (see answer to the first comment).

RC> l 553 Insert 'the' in front of majority.

AR> We will insert it in the revised manuscript.

RC> l 559 Remove 'to' in from of streams.

AR> We will remove it in the revised manuscript.

RC> Table 1 call this soil physicochemical parameters as it's not really biochemical.

AR> We agree with this comment and we will change it accordingly.

RC> Table 4 Did you quote the decimals to significant figures?

AR> We reported the values to their significant digit based on their standard error.

RC> Figures 2,3 and 5 either call the x axis 'month' or remove 'months' all together as it is obvious that is a data axis (just add years).

AR> We will modify the pictures by removing "months" and leaving just the years in the x-axis.

RC> Figure 3 are RCOO and HCO3 combined or separate? Please clarify. If it is combing perhaps say '+'?

AR> We prefer to not modify this in the picture but we will clarify that it is the combined contribution in the picture description (as it will be done in the text in accordance with the comment above).

RC> l 1010 with 'unpublished' do you mean not yet published? If it is in prep, please add, otherwise remove this citation. You are contracting 'reported' with 'unpublished'.

AR>The data are reported as unpublished because they are presented as original data in a manuscript currently under review in another journal. To avoid confusion we will modify the note as: "These data are not included in the manuscript to avoid double-publication as they were already included in another manuscript not yet published".

[Figure]

[Figure]

**Fig. 1.** Figure A2 Annual yield for each experimental treatment from 2017 to 2019.

---

## Author Comment (AC2) · 10 Aug 2020

RC> Abstract: It would be interesting to indicate some values of nutrient losses in plots with conventional management practices.

AR> We agree with the reviewer and decided to add the values in l 28 as follows: "The leaching of total N was the highest with conventional management (73.6 kg N ha-1 yr-1) and the lowest in mechanical weeding with reduced fertilization (32.0 kg N ha-1 yr-1) whereas its yield remained comparable among all treatments".

RC> L32: "Our findings signified that mechanical weeding: : :" should be replaced by

"Our findings suggested that mechanical weeding: : :" because you cannot generalize to Indonesia your results at a single site.

AR> We agree with this comment and we will modify it accordingly.

RC> L71-73: Not always the case, see for example eucalypt plantations.

AR> We agreed that this is not always the case and therefore we will substitute "result" with "can result". See below for a literature review on eucalyptus plantations.

RC> The lysimetry design was suitable for the quantification of leaching losses. You might indicate that, even though the time period for soil stabilization was short in your study (only two months from the installation of the ceramic cups to the start of the soil solution collection and four months from the implementation of the factorial management experiment to the collection of the soil solutions), this period was sufficient because the biological processes are rapid in tropical soils.

AR> Thank you for this observation, indeed the period was quite short but it was enough to show effects on nutrient leaching. However, other soil parameters (e.g. gross N mineralization, soil nutrient contents) did not yet show any treatment effect, probably due to the short period passed between the establishment of the experiment and sampling. Therefore we prefer to not add this in the text to avoid confusion.

RC> You may be interested in two articles that accurately describe the spatial development of roots in oil palm plantations: Plant and Soil 189: 33-48, 1997 and Plant and Soil 190: 235-246, 1997.

AR> Thank you for this comment, I am familiar with these articles but I was not sure where to insert them in the paper. I will insert at l 97 a small sentence to describe root density of the oil palm: "Root uptake is related to root density, which is high inside the palm circle and lower in the inter-row (Jourdan and Rey, 1997; Lamade et al. 1996)" [Jourdan, C. and Rey, H.: Plant and Soil, 190, 235–246, 1997].

RC> More information should be given on the water drainage model. How have you

dealt with run off from one management zone to another?

AR> It's not possible to include this in the water model because the different management zones have to be modelled separately, with no interaction with each other. Nevertheless, we expect the runoff to be small in our plantation based on literature review in oil palm plantations and the lack of slope at our site.

RC> The parameterization of the model is rough, without measurements of root profiles and soil hydraulic parameters in each treatment and management zones. Moreover, the validation from soil water potentials is also rough with only two depths (it would have been interesting to include the depth of 1.5 m where soil solutions are collected) and punctual measurements in only two treatments (only 12 tensiometers). It is important to provide a table (in appendix) showing the values given to all the parameters used in the Expert-N water sub-model for each management zone.

AR> We agree that the parametrization of the water model is rough because of the reasons explained. Unfortunately, we could not find an easily accessible, spatially explicit water model that could account for the spatial variability given by the management zones. The commonly used models estimate root water uptake from an estimation of plant evapotranspiration and root density, and therefore cannot partition the root water uptake among management zones. On the other hand, more complicated models require in-depth knowledge of the processes and a large quantity of data that are often not available in literature. Given the limits in our model parameterization, our modeling approach strongly relied on the calibration of the results with field measurements of soil matric potential. We focused on the top 60 cm of the soil because the majority of roots in oil palm are in the top-50-cm depth, so that this is the main zone where the water is exchanged between the soil and the plant and with the atmosphere. The table with all the parameters used in the Expert-N sub-model will be provided in the appendix as Table A1.

RC> Discussion You might be interested by a recent paper providing values for N

leaching in oil palm plantations. The three management zones were sampled in the field to validate this model and the order of magnitude is consistent with your results: DOI:10.1002/agj2.20109.

AR> Thank you for providing this important reference. I will include that in the manuscript.

RC> The comparison with other perennial tropical plantations is interesting (Table A2). Could you add forest plantations to this appendix (pines, eucalypts, acacias) and rubber plantations if you find data in the literature?

AR> Nutrient leaching losses in rubber smallholder plantations were measured near our study site by our group: annual N leaching losses were 4 kg N ha-1 yr-1 (Kurniawan et al. 2018). These data were not included in the table because these plantations were not fertilized. We couldn't find other data about field-measured leaching losses in rubber plantations or other plantations, just a few data in eucalyptus plantations. Silva et al. 2013 (Forest Ecology and Management 301: 67-78) measured leaching in 2-years-old eucalyptus plantations on sandy soil in Brazil with low fertilization (80 kg N ha-1 in 2 years) and lower annual rainfall than our site (1240 mm). They found relatively low leaching: 5.6 kg N ha-1 leached in 2 years when the fertilizer was applied 4 times over the 2-year study period, and 8.6 kg N ha-1 leached in 2 years when the fertilizer was applied one time at the beginning of the sampling. Another interesting study on nutrient dynamics in eucalyptus plantation is the one of Laclau et al. 2010 (Forest Ecology and Management 259(9):1771-1785) for plantations of different ages in Brazil and Congo, fertilized once with 38 kg N ha-1 (Congo) and 120 kg N ha-1 (Brazil). The leaching fluxes were in the range of 1-6 kg N ha-1 yr-1. Another study by this group, published by Versini et al. 2014 (Geoderma 232-243: 426-436), on 2-years-old eucalyptus plantations in Congo (annual rainfall of 1220 mm and fertilization of 43 kg N ha-1 at planting), measured similar leaching losses, equal to 4.4 kg N ha-1 yr-1. We decided to not include these data on eucalyptus plantations in Table A2 because: 1) the rainfall was much lower than the one at our site 2) the majority of the data were

from young plantations, 3) the plantations were not regularly fertilized.

Silva et al. 2013, soil:Arenosols, precipitation: 1240 mm yr-1, plantation: 2-years-old eucalyptus, N fertilizer: 40 kg N ha-1 yr-1, N leaching: 7 kg N ha-1 yr-1, % fertilizer leached: 17.5%

Versini et al. 2014, soil:Arenosols, precipitation: 1220 mm yr-1, plantation: 2-years-old eucalyptus, N fertilizer: 21.5 kg N ha-1 yr-1, N leaching: 4.4 kg N ha-1 yr-1, % fertilizer leached: 20.5%.

RC> L515-517: not sure that the amounts of Na taken up by soil macrofauna could be sufficient to explain the differences. It has been demonstrated that oil palm can take up Na in addition/substitution to K (Bonneau, X., Boutin, D., Bourgoing, R., Sugarianto, J., 1997. Le chlorure de sodium, fertilisant idéal du cocotier en Indonésie. Plantations, Recherche, Développement 4 (5), 336–346), as also shown recently in eucalypt plantations.

AR> Thank you for providing these interesting references that show that palm and trees in plantations can take up Na from the soil. However, we think that this cannot explain lower Na leaching with mechanical weeding compared to herbicide weeding because it would imply higher Na uptake by the palms with mechanical weeding, which was not investigated.

RC> L329: than in the inter-row.

AR> We will add "in" in the revised manuscript.

RC> L332: dissolved organic N or total dissolved N?

AR> We understand that this may be misleading and we decided to remove "dissolved" in l 332.

RC> L453: higher?

AR> We will change "highest" with "higher" in the revised manuscript.

Please also note the supplement to this comment:
https://bg.copernicus.org/preprints/bg-2020-153/bg-2020-153-AC2-supplement.pdf

**Supplement:**

**Table A1** Parameters used for the Expert-N water sub-model for each management zone

| | Depth (cm) | Palm circle | Inter-row | Frond-stacked area |
|---|---|---|---|---|
| **Interception** | | | | |
| Saturation capacity (mm d$^{-1}$) | | 8.4 | 4.7 | 4.7 |
| Throughfall (%) | | 50 | 10 | 10 |
| **Plant water uptake** | | | | |
| Plant height (cm) | | 874 | 874 | 874 |
| Leaf area index | | 3.64 | 1.8 | 0.75 |
| Leaf number | | 40 | 40 | 40 |
| Aboveground biomass (kg ha$^{-1}$) | | 47400 | 47400 | 47400 |
| Maximum rooting depth (cm) | | 100 | 50 | 50 |
| Root biomass (kg ha$^{-1}$) | | 15600 | 15600 | 15600 |
| Root partition (%) | 0–10 | 29 | 29 | 29 |
| | 10–30 | 31 | 31 | 31 |
| | 30–50 | 18 | 18 | 18 |
| | 50–100 | 15 | 15 | 15 |
| | 100–150 | 5 | 5 | 5 |
| | 150–200 | 2 | 2 | 2 |
| **Soil properties** | | | | |
| Bulk density (g cm$^{-3}$) | 0–10 | 1.37 | 1.36 | 0.8 |
| | 10–30 | 1.36 | 1.36 | 1.26 |
| | 30–50 | 1.52 | 1.52 | 1.52 |
| | 50–100 | 1.50 | 1.50 | 1.50 |
| | 100–150 | 1.58 | 1.58 | 1.58 |
| | 150–200 | 1.46 | 1.46 | 1.46 |
| Texture – Clay (%) | 0–10 | 15.8 | 15.8 | 15.8 |
| | 10–30 | 24.5 | 24.5 | 24.5 |
| | 30–50 | 37.5 | 37.5 | 37.5 |
| | 50–100 | 41.0 | 41.0 | 41.0 |
| | 100–150 | 43.3 | 43.3 | 43.3 |
| | 150–200 | 47.6 | 47.6 | 47.6 |
| Texture – Sand (%) | 0–10 | 53.3 | 53.3 | 53.3 |
| | 10–30 | 47.6 | 47.6 | 47.6 |
| | 30–50 | 35.9 | 35.9 | 35.9 |
| | 50–100 | 34.4 | 34.4 | 34.4 |
| | 100–150 | 31.7 | 31.7 | 31.7 |
| | 150–200 | 29.8 | 29.8 | 29.8 |
| Organic matter (%) | 0–10 | 3.2 | 2.9 | 8.7 |
| | 10–30 | 2.8 | 2.6 | 3.7 |

| | | | | |
|---|---|---|---|---|
| | 30–50 | 2.0 | 1.6 | 2.0 |
| | 50–100 | 2.5 | 2.5 | 2.5 |
| | 100–150 | 2.0 | 2.0 | 2.0 |
| | 150–200 | 1.2 | 1.2 | 1.2 |
| Porosity (Vol %) | 0–10 | 48.8 | 48.8 | 70.0 |
| | 10–30 | 45.7 | 45.7 | 45.7 |
| | 30–50 | 41.9 | 41.9 | 41.9 |
| | 50–100 | 43.3 | 43.3 | 43.3 |
| | 100–150 | 40.3 | 40.3 | 40.3 |
| | 150–200 | 45.0 | 45.0 | 45.0 |
| Field capacity (Vol %) | 0–10 | 27.2 | 27.2 | 27.2 |
| | 10–30 | 27.4 | 27.4 | 27.4 |
| | 30–50 | 21.3 | 21.3 | 21.3 |
| | 50–100 | 23.1 | 23.1 | 23.1 |
| | 100–150 | 24.5 | 24.5 | 24.5 |
| | 150–200 | 28.1 | 28.1 | 28.1 |
| Wilting point (Vol %) | 0–10 | 18.3 | 18.3 | 18.3 |
| | 10–30 | 17.3 | 17.3 | 17.3 |
| | 30–50 | 17.9 | 17.9 | 17.9 |
| | 50–100 | 17.3 | 17.3 | 17.3 |
| | 100–150 | 20.4 | 20.4 | 20.4 |
| | 150–200 | 24.5 | 24.5 | 24.5 |
| Saturated hydraulic conductivity (mm d$^{-1}$) | 0–10 | 400 | 400 | 200 |
| | 10–30 | 200 | 200 | 400 |
| | 30–50 | 200 | 200 | 300 |
| | 50–100 | 150 | 150 | 150 |
| | 100–150 | 260 | 260 | 260 |
| | 150–200 | 260 | 260 | 260 |
| Van Genuchten $\alpha$ (cm$^{-1}$) | 0–10 | 0.059 | 0.059 | 0.059 |
| | 10–30 | 0.025 | 0.025 | 0.035 |
| | 30–50 | 0.010 | 0.010 | 0.020 |
| | 50–100 | 0.008 | 0.008 | 0.015 |
| | 100–150 | 0.021 | 0.021 | 0.021 |
| | 150–200 | 0.021 | 0.021 | 0.021 |
| Van Genuchten n | 0–10 | 1.70 | 1.70 | 1.70 |
| | 10–30 | 1.71 | 1.71 | 1.81 |
| | 30–50 | 1.12 | 1.12 | 1.25 |
| | 50–100 | 1.09 | 1.09 | 1.15 |
| | 100–150 | 1.21 | 1.21 | 1.21 |
| | 150–200 | 1.23 | 1.23 | 1.23 |

---

## Author Comment (AC3) · 10 Aug 2020

RC> Line 23-28 Could you consider the specific data (e.g., low solute concentrations, small drainage: : :) should be added, and thus increasing the persuasiveness for the readers? (Maybe it is important in the Abstract).

AR> We agree that is important to insert some specific data but we want to avoid getting a long abstract. We think that the difference in leaching losses between conventional and reduced management are the most important numbers to be shown so we will include them in the abstract by revising l 27-31 as follows: "Mechanical weeding reduced leaching losses of all nutrients compared to the conventional herbicide

weeding, because herbicide decreased ground vegetation, and thereby reduced the efficiency of soil nutrient retention. The leaching of total N was the highest with conventional management (73.6 kg N ha-1 yr-1) and the lowest in mechanical weeding with reduced fertilization (32.0 kg N ha-1 yr-1) whereas its yield remained comparable among all treatments".

RC> 1. Line 41 The term "e.g" should be "e.g.".

AR> We will correct the punctuation in the revised manuscript.

RC> Line 52-54 Could you provide some references? Thank you.

AR> Since we don't have strong evidence to justify this sentence, we will restructure it in the following way: "The decline in soil fertility reinforces the dependency on fertilizer inputs and threatens the long-term productivity of the area (Syers 1997), possibly exacerbating land-use change". [Phil. Trans. R. Soc. Lond. B. 352 (1356) :1011-1021]

RC> Line 57-63 Indeed, high precipitation rate is a critical driver for surface runoff and associated nutrient losses. Particularly for considerable plantations. However, the leaching losses may be offset by a high nutrient cycling due to the rapid uptake of plants.

AR> Thank you for this observation; this was expanded in the manuscript in the second paragraph.

RC> Line 92 How far the radius of the palm circle?

AR> We will include it in the revised manuscript.

RC> Line 98-99 Could you describe more details about the root distribution of oil palm? I am not sure the roots of palm only grow around the trunk. I think the root biomass between inter-row area may be high in somewhere.

AR> The root density is higher in the palm circle because it's closer to the palm stem and because of the repetitive fertilization in this area, whereas the inter-row has the

lower root density. We will include a sentence at l 97 in the revised manuscript: "Root uptake is related to root density, which is high inside the palm circle and lower in the inter-row (Jourdan and Rey, 1997; Lamade et al. 1996)". [Jourdan, C. and Rey, H.: Plant and Soil, 190, 235–246, 1997].

RC> Line 159 Replace the "x" between "the 50 x 50 m" with "_".

AR> We prefer to not modify this because the symbol "_" may cause confusion to the reader since we want to indicate a mathematical product, normally identified by the symbol "x".

RC> Line 160 Where is the plant materials from the mechanical weeding? Are they transported far away from the plots?

AR> The plant material is left inside the plot. We will add this detail to the text.

RC> Line 232 Is the runoff set to 0? Do you mean "no overland runoff"?

AR> Yes, we meant this, we will specify no overland runoff in brackets in the revised manuscript.

RC> Line 243 Soil physical-chemical characteristics.

AR> We will modify this in the revised manuscript.

RC> Line 247 See comment 1.

AR> Please see the answer to comment 1.

RC> Line 265 Please simplify the statistical analyses section.

AR> Although we understand that the statistical analyses section may seem too long, we cannot simplify it without removing essential information. We think that is in line with good scientific practices to report all the statistical analyses used, for reproducibility.

RC> The results section is well-organized manner. However, some statements are so long that they (e.g., the section "3.2 Differences in leaching losses: : :") should be

simplified to delete some non-key contents.

AR> We decided to include some explanation sentences in the results to guide the reader since we presented a lot of results. Following this comment, we carefully re-examined the manuscript and decided to modify the l 343 as follows: "On the other hand, base cation leaching fluxes had opposite patterns as their concentrations as Ca, K, and Mg leaching were higher in the frond-stacked area than the palm circle and inter-row (all P < 0.01; Fig. 4). Leaching of Na was higher in both the frond-stacked area and inter-row than the palm circle (P < 0.01; Fig. 4)".

RC> Line 310-311 The drainage flux is low. Do you investigate the stem flow (may be influenced by "funnel effect" of canopy of oil palm?)? Some studies demonstrated that the infiltration was enhanced around the tree trunk.

AR> The model used did not allow to parametrize the stem flow and the funnel effect. Nevertheless, the values from the water model were calibrated with field measurements of soil matric potential, reflecting the actual field conditions.

RC> Line 346 Why is different between the various elements leaching?

AR> The differences in nutrient leaching fluxes between management zones depend on 1) differences in water drainage fluxes (the same of every element) and 2) differences in element concentrations in deep soil-water (which varies for each element). For example, much higher Al concentrations in the inter-row compared to the frond-stacked area resulted in higher Al leaching even if the frond-stacked area have higher water drainage. On the other hand, Mg concentrations were comparable among inter-row and frond-stacked area, but due to the large drainage of the latter Mg leaching was higher in the frond-stacked area.

RC> Line 391 I recommend the ratios of runoff/interception/evaporation/transpiration to precipitation was supplemented in the Table 2 for better understanding.

AR> We would prefer to not include that in Table 2 to have a simpler table and because

the ratios can easily be extrapolated. However, we will include the ratio if requested by the editor.

RC> Line 434-438 How to understand the use of organic amendments and slow-release fertilizers? E.g., mulching application? Under the high temperature and precipitation in some tropical areas, the plant materials decompose quickly, and the litterfall may have very short residence time on the ground. Could you provide any information on the standing plant litter in your treatments? Thank you!

AR> In the studied plantation, there were two peaks of leaching due to the overlapping of high drainage fluxes and fertilizer application. Since it is complicated to predict the periods with high drainage, it may be useful to use organic or slow-release fertilizers. These can distribute the nutrient input to the soil over a longer period of time, thus reducing the overlapping of high nutrient input and high drainage, and also avoiding peaks of high nutrient inputs. Indeed, the decomposition in the tropics happens fast but it would still provide nutrients slower than mineral fertilization. The organic amendments more used in oil palm plantations are the waste from the palm oil processing, namely the empty fruit bunches (EFB) or the palm oil mill effluent (but this is normally applyied just close to the mill). In a study on decomposition rates and nutrient release by EFB in oil palm plantation, 75% of the EFB was decomposed in 8 months, with a deposition constant k= 0.2 month-1 (Moradi et al. 2014, Ann Appl Biol 164: 208–219). Another organic amendment in the oil palm plantation is the litterfall, which is represented by the frond stack. This can provide plenty of nutrients to the plantation as it can be seen from the high nutrient contents in the frond-stacked area in our manuscript (Table 1). Also, the addition of cut fronds is regular, at a rate of 16 fronds palm-1 yr-1 in the studied plantation. However, this positive effect on soil fertility is restricted to the frond-stacked area and it is unsure if the palm can benefit from these nutrients. Recent literature (Rüegg et al., 2019) found higher root density under the frond-stacked area compared to the inter-row, indicating that the palm may indeed take up the nutrients from the decomposition of the frond stack.

RC> Line 552-553 Although the mechanical weeding is sustainable way in ecological view, the farmers were reluctant to adopt due to its money-consuming and labor-consuming. Undoubtedly, mechanical weeding is a promising measure to reduce nutrient leaching.

AR> Indeed we would expect the farmers to be reluctant to adopt this weeding method because it requires more labor. However, the economic analysis done in Darras et al. (2019) showed that the costs to implement mechanical weeding would be comparable to the ones for herbicide application. In fact, the prolonged use of glyphosate as standard weeding practice has favored woody and resistant understory vegetation that have to be cut periodically with mechanical weeding.

---

## Author Response (AR1)

Dear Dr. Sara Vicca, on behalf of my co-authors, I express my sincere gratitude for the helpful reviews and comments on our manuscript **bg-2020-15**. We have now incorporated all the changes we stipulated in our answers to the reviewers' comments and from your suggestions. All the line numbers are based on the revised manuscript (not on the marked-up version where the line numbers change).

We hope that our revisions will satisfy your and the reviewers' questions and the standards of Biogeosciences. We look forward to hearing back from you. If there are any questions regarding our manuscript, I would be happy to clarify.

Sincerely yours,
Greta Formaglio

**Comments from the editor:**

Regarding the short time for the soil to adjust after lysimeters were installed, I would like you to include a brief note about the suitability for nutrient leaching but not for some other soil processes. This to make readers aware and cautious about it when they would conduct similar experiments.

Author's response: in addition to fast cycling in the tropics, the short time period to adjust was also justified by the minimal soil disturbance during the installation of the lysimeter, as we used an auger with the approximate same diameter as the installed lysimeter.

Author's changes in the manuscript: we added this at L 177.

In the Discussion, you mention that fertilization should be avoided during periods of high drainage fluxes. Based on the referee comments, you suggest to modify this to indicate that fertilization during the period of high drainage fluxes should be reduced. Could it be an option to (also) spread the fertilization a bit more in time during that period?

Author's response: yes, indeed that is an option.
Author's changes in the manuscript: we added this at L 432.

I also like to repeat the request by referee 1 to carefully check spelling and grammar.

Author's response: we checked the manuscript thoroughly.

Author's changes in the manuscript: we corrected Table A2, in which the gross N mineralization rates were reported on mass basis (mg N $kg^{-1}$ $d^{-1}$), instead of being on area-based (mg N $m^{-2}$ $d^{-1}$).

**Comments from Reviewer 1:**

My main comment is that the analysis/results of yield are not appropriately considered. The authors refer to previously published studies on the yield aspect but do not actually quote any data/numbers on yield. This is an important point that is missing. Any management options leading to more environmentally friendly or sustainable growing of oil palm will ultimately be scrutinised under the yield aspect. So it is important to include the actual figures in this assessment (even if they have been published elsewhere). Just saying there was no difference is not sufficient. Growers would like to know the actual yield to see that you were not working on a plantation with unusually high or low yield anyway which might have masked any management effects. So a revised version needs results and discussion sections that are expanded with details on yield.

Author's response: thank you for this observation. We agreed that the focus on the yield is essential for the development of long-term more sustainable management practices. The average yield measured was in the same range as the yield reported for large-scale plantations in Indonesia.

Author's changes in the manuscript: Fig. A2 was included and a sentence about the yield was added at L520-525.

General:

There should be no space between number and %. Please revise this throughout the manuscript.

Author's response: corrected.

Author's changes in the manuscript: L45, 138, 139, 152, 153, 219, 227, 230, 281, 282, 316, 318, 336, 337, 363, 364, 365, 366, 367, 394, 395, 539, 540, 566, 567, 993.

The term 'stem' might be more appropriate to use than 'trunk'. Please replace 'trunk' with 'stem' throughout the manuscript.

Author's response: we agree.

Author's changes in the manuscript: changed at L 93, 138, 169, 170, 227, 460.

The term 'conventional' is a bit misleading. Perhaps 'standard practice' or 'standard industry practice' would be a more appropriate term to use?

Author's response: we prefer to keep the term of conventional practice for consistency with other manuscripts published on this experiment, i.e. Darras et al. 2019, and others in preparation.

Author's changes in the manuscript: no change.

Generally spelling and grammar need to be checked carefully, some sentences are too long and convoluted.

Author's response: based on this comment we revised the manuscript and decide to modify the structure of a few sentences to improve clarity.

Author's changes in the manuscript: some sentence were modified L 66-67, 346-349, 395-400, 411-416, 443-445, 462-465, 491-493.

Specific:

Title remove 'as'.

Author's response: this is taken out

Author's changes in the manuscript: L1.

l 39 replace 'have' with 'has'.

Author's response: corrected.

Author's changes in the manuscript: L39.

l 40 remove 'and'.

Author's response: corrected.

Author's changes in the manuscript: L 40.

l 45 remove space between 57 and % (see general comment above).

Author's response: corrected.

Author's changes in the manuscript: see general comment above.

l 48 introduce N as nitrogen.

l 59 introduce P as phosphor $NO_3$ as nitrate.

Author's response: we introduced "nitrate" but not the common elements, since N, P, K etc. are all generally known elemental abbreviations.

Author's changes in the manuscript: L 59.

l 61 comma after reference before 'whereas'.

Author's response: corrected.

Author's changes in the manuscript: L60.

l 68 remove 'of' in front of oil palm.

Author's response: corrected.

Author's changes in the manuscript: L 66-67 were revised.

l 80 herbicides needs to be plural.

Author's response: corrected.

Author's changes in the manuscript: L79.

l 82 'herbicide weeding' perhaps clarify that this is chemical weeding with herbicides as supposed to mechanical weeding.

Author's response: in this case we meant chemical weeding.

Author's changes in the manuscript: specified that is chemical weeding at L 81.

l 97 circles needs to be plural.

Author's response: we prefer to keep it singular for consistency, as it is the management zone mentioned above.

Author's changes in the manuscript: no change.

l 101 canopy interception will depend on the age of the plantation and whether there is canopy closure or not. Please elaborate here and say that there will be a difference between younger and older plantations.

Author's response: we agree that the age of the palm is important to determine the level of interception in the inter-row. However, we think that this sentence doesn´t need to be modified because the interception in the inter-row will always be lower than in the palm circle, independent of the age of the palm. Even when the canopy closes, 7-8 years after planting, the interception is lower in the inter-row than in the palm circle (Banabas et al. 2008).

Author's changes in the manuscript: no change.

l 131 Was the plantation terraced? This should be a discussion point generally as nutrient flows will be different in plantations with even terrain as supposed to terraced plantations. It seems to become more popular (also in Indonesia) to terrace plantations (when replanting) even if the terrain is not that hilly to begin with. This might have potentially large implications on nutrient leaching.

Author's response: the plantation was not terraced and it was not on a slope position.

Author's changes in the manuscript: a sentence added at L 134.

l 135 remove space between number and %.

Author's response: corrected

Author's changes in the manuscript: see general comment above.

l 137 chemically or mechanically weeded? Please clarify.

Author's response: this was already specified in the introduction.

Author's changes in the manuscript: we specified that it was chemically wedded at L 139.

l 142 Is the fertiliser applied in pellets or granules? Broad spread? Please give details.

Author's response: the fertilizers were in granular form and these were banded application within the palm circle, not broadcasted.

Author's changes in the manuscript: specified in L 148, 150.

l 145 Were no EFB (empty fruit bunches) returned to the plantation? This is common practice and would add more organic matter to the plantation in addition to the palm fronds. If it wasn't done in your plantation, it still needs to be discussed in the discussion section.

Author's response: in our studied plantation there is a certain amount of EFB returned to the plantation (mainly in the form of compost). The 2025-ha plantation investigated is owned by the company PTPN6, which has a total of 90122 ha of oil palm plantations in Jambi province. Once the fruits are processed in the mill, the EFB (also in the form of compost or ashes) is redistributed across all the plantations of PTPN6 in a rotation. In addition, within the same plantation, the application of EFB follows a rotation system based on management blocks. Therefore the timing and the amount of EFB distributed are quite complicated to predict. In our experimental plots, we did not include the EFB compost in the treatments. This is for two reasons: 1) our experimental plots encompassed different management blocks and therefore the different timing and irregular application of EFB might have biased the plots; 2) the aim of the experiment is to compare the most standardized management practices with the reduced management intensity, and the application of EFB is not done regularly throughout the plantations (it is normally applied just to the area next to the mill, for ease of transport by the workers). Since we didn´t include the practice of using EFB in our experiment, we cannot discuss its effects on leaching, but we included this for future systematic evaluation.

Author's changes in the manuscript: EFB added at L 434-437.

l 170 replace 'till' with 'to'.

Author's response: corrected.

Author's changes in the manuscript: L 172.

l 182 check reference, should only be (2018) in brackets.

Author's response: corrected.

Author's changes in the manuscript: L187.

l 191 not clear what (2018) is suppose to mean? Should there be a reference? And do you mean combined bicarbonate and organic acids? Please clarify (also in Fig 3).

Author's response: yes, the (2018) was from the previous study by Kurniawan et al. (2018).

Author's changes in the manuscript: reference corrected at L 195 and the word "combined" added at L 194 to avoid confusion.

l 205 do you have anything to base your assumption on? Any measurements/references?

Author's response: we don´t have strong references to back down this sentence, therefore we prefer to remove the last part of the sentence

Author's changes in the manuscript: L 207.

l 209 insert 'The' in front of Expert-N model.

Author's response: corrected.

Author's changes in the manuscript: 212.

l 211 insert 'the' after using.

Author's response: corrected.

Author's changes in the manuscript: L 214.

l 217 add reference for Richards' equation.

Author's response: added.

Author's changes in the manuscript: L 220.

l 224 'stem' instead of 'trunk'.

Author's response: changed

Author's changes in the manuscript: see general comment above.

l 237 space between 1 and um.

Author's response: corrected

Author's changes in the manuscript: L 240.

l 246 remove 'to' in front of 192.

Author's response: corrected

Author's changes in the manuscript: L 249.

l 306 replace 'high' with 'strong'.

Author's response: corrected

Author's changes in the manuscript: L309.

l 318 specify here the months for typical dry and wet season.

Author's response: we think that specifying the typical months will be confusing. The dry period is clearly indicated in the picture that is referred in the text.

Author's changes in the manuscript: no changes.

l 424 But as you are saying elsewhere, they would not fertilise in the dry season either. So you might have to elaborate here.

Author's response: it is true that the farmers don´t fertilise in the dry season, which is also not advisable for the reasons explained in l 429 but they could fertilize for example one month later, avoiding the peaks of drainage. Unfortunately, as explained in the manuscript, this is not practical because it is difficult to predict the period of high drainage fluxes from the precipitation pattern and so the most viable option to reduce leaching is to reduce fertilization rates and increase nutrient retention efficiency.

Author's changes in the manuscript: no changes.

l 438 Mention EFB, POME, compost or a combination of these as they are all commonly used types of organic fertiliser in oil palm plantations.

Author's response: this is now added.

Author's changes in the manuscript: L 434-437.

l 443 circle closest to the roots for direct uptake of the fertilisers?

Author's response: this was explained in detail later in the manuscript (L 469-470).

Author's changes in the manuscript: no changes.

l 453 replace 'highest' with 'higher'.

Author's response: we replaced with "larger".

Author's changes in the manuscript: L 455.

l 458 trunk = stem

Author's response: corrected.

Author's changes in the manuscript: see general comment above.

l 464 insert 'will' in front of largely.

Author's response: corrected.

Author's changes in the manuscript: L 466.

l 492 replace 'increased' with 'increases'.

Author's response: corrected.

Author's changes in the manuscript: L 494.

l 494 The last part of the sentence doesn't fit the first. Please rephrase.

Author's response: the last part of the sentence was removed to improve clarity, as the study is not investigating soil fertility.

Author's changes in the manuscript: L 501.

l 502 replace 'influence' with 'effect'.

Author's response: corrected.

Author's changes in the manuscript: L 504.

l 505 Please rephrase this sentence, it sounds a bit clumsy and i is not that clear what you are trying to say.

Author's response: revised as suggested.

Author's changes in the manuscript: L 505-508.

l 508 Replace 'have' with 'has'.

Author's response: corrected.

Author's changes in the manuscript: L 510.

l 519 Start a new sentence after the reference as the second half doesn't follow the first.

Author's response: revised as suggested.

Author's changes in the manuscript: L 528.

l 525 replace 'conventional' with 'standard practice'?

Author's response: we prefer to keep the world "conventional" for consistency throughout the manuscript.

Author's changes in the manuscript: no changes.

l 527 This is where you need to expand on the yield aspect and include data etc.

Author's response:  we decided that it is to include the yield in L 520-522 where we first mentioned the economic aspects of the treatments (see answer to the first comment above).

Author's changes in the manuscript: L 520-522.

l 553 Insert 'the' in front of majority.

Author's response: corrected.

Author's changes in the manuscript: L 562.

l 559 Remove 'to' in from of streams.

Author's response: corrected.

Author's changes in the manuscript: L568.

Table 1 call this soil physicochemical parameters as it's not really biochemical.

Author's response: SOC, total N, [15]N natural abundance are the biochemical characteristics, which include the other chemical parameters.

Author's changes in the manuscript: no changes.

Table 4 Did you quote the decimals to significant figures?

Author's response: the difference in values are minute and thus the values (mean $\pm$ se) must appropriately have 3 decimal places. If expressed in percentage, this will have 1 decimal place, which is acceptable.

Author's changes in the manuscript: no changes.

Figures 2,3 and 5 either call the x axis 'month' or remove 'months' all together as it is obvious that is a data axis (just add years).

Author's response: "months" removed from the pictures

Author's changes in the manuscript: Fig. 1 (L 1003), Fig. 2 (L 1007), Fig. 4 (L 1019).

Figure 3 are RCOO and HCO3 combined or separate? Please clarify. If it is combing perhaps say '+'?

Author's response: we prefer to not modify this in the fig. but this is now specified in the fig. caption that this is a combined contribution $RCOO^-$ and $HCO_3^-$.

Author's changes in the manuscript: L 1010.

l 1010 with 'unpublished' do you mean not yet published? If it is in prep, please add, otherwise remove this citation. You are contracting 'reported' with 'unpublished'.

Author's response: note revised.

Author's changes in the manuscript: L 1037-1038.

**Comments from Reviewer 2:**

Abstract:

It would be interesting to indicate some values of nutrient losses in plots with conventional management practices.

Author's response: we agree with this comment and we added some values.

Author's changes in the manuscript: L 27-32 were revised.

L32: "Our findings signified that mechanical weeding…" should be replaced by "Our findings suggested that mechanical weeding…" because you cannot generalize to Indonesia your results at a single site.

Author's response: we agree with this comment.

Author's changes in the manuscript: we modified "signified" with "suggested" L 32.

Introduction: The Introduction section is clear and informative.

L71-73: Not always the case, see for example eucalypt plantations.

Author's response: we agree that this is not always the case and therefore we will substitute "result" with "can result". See below for a literature review on eucalyptus plantations.

Author's changes in the manuscript: L 72.

Material and methods

The lysimetry design was suitable for the quantification of leaching losses. You might indicate that, even though the time period for soil stabilization was short in your study (only two months from the installation of the ceramic cups to the start of the soil solution collection and four months from the implementation of the factorial management experiment to the collection of the soil solutions), this period was sufficient because the biological processes are rapid in tropical soils.

Author's response: thank you for this observation, indeed the period was quite short but it was enough to show effects on nutrient leaching, also because the soil disturbance was minimized by using an auger with similar diameter as the installed lysimeter.

Author's changes in the manuscript: this was inserted at L 177-179.

You may be interested in two articles that accurately describe the spatial development of roots in oil palm plantations: Plant and Soil 189: 33-48, 1997 and Plant and Soil 190: 235-246, 1997.

Author's response: thank you for this comment, we are familiar with these articles. We agree to include one as a reference in the manuscript.
Author's changes in the manuscript: a sentence to describe root density was inserted at L 97-98.

More information should be given on the water drainage model. How have you dealt with run off from one management zone to another?

Author's response: it´s not possible to include this in the water model because the different management zones have to be modelled separately, with no interaction with each other. Nevertheless, we expect the runoff to be small in our plantation based on literature review in oil palm plantations and the lack of slope at our site.

Author's changes in the manuscript: we inserted a table describing all the parameter use to model water drainage (Table A1, see comment below).

The parameterization of the model is rough, without measurements of root profiles and soil hydraulic parameters in each treatment and management zones. Moreover, the validation from soil water potentials is also rough with only two depths (it would have been interesting to include the depth of 1.5 m where soil solutions are collected) and punctual measurements in only two treatments (only 12 tensiometers). It is important to provide a table (in appendix) showing the values given to all the parameters used in the Expert-N water sub-model for each management zone.

Author's response: we agree that the parametrization of the water model is rough because of the reasons explained. Unfortunately, we could not find an easily accessible, spatially explicit water model that could account for the spatial variability given by the management zones. The commonly used models estimate root water uptake from an estimation of plant evapotranspiration and root density, and therefore cannot partition the root water uptake among management zones. On the other hand, more complicated models require in-depth knowledge of the processes and a large quantity of data that are often not available in literature. Given the limits in our model parameterization, our modeling approach strongly relied on the calibration of the results with field measurements of soil matric potential. We focused on the top 60 cm of the soil because the majority of roots in oil palm are in the top-50-cm depth, so that this is the main zone where the water is exchanged between the soil and the plant and with the atmosphere. Author's changes in the manuscript: the table with all the parameters used in the Expert-N sub-model will be provided in the appendix as Table A1 (L 1031).

Statistical analyses are clearly presented.

Results No specific recommendation, this section is well written.

Discussion

You might be interested by a recent paper providing values for N leaching in oil palm plantations. The three management zones were sampled in the field to validate this model and the order of magnitude is consistent with your results: DOI:10.1002/agj2.20109. Author's response: thank you for providing this important reference. I will include that in the manuscript.
Author's changes in the manuscript: reference inserted at L 558-560.

The comparison with other perennial tropical plantations is interesting (Table A2). Could you add forest plantations to this appendix (pines, eucalypts, acacias) and rubber plantations if you find data in the literature?

Author's response: nutrient leaching losses in rubber smallholder plantations were measured near our study site by our group: annual N leaching losses were 4 kg N ha$^{-1}$ yr$^{-1}$ (Kurniawan et al. 2018). These data were not included in the table because these plantations were not fertilized. We couldn´t find other data about field-measured leaching losses in rubber plantations or other plantations, just a few data in eucalyptus plantations. Silva et al. 2013 (Forest Ecology and Management 301: 67-78) measured leaching in 2-years-old eucalyptus plantations on sandy soil in Brazil with low fertilization (80 kg N ha$^{-1}$ in 2 years) and lower annual rainfall than our site (1240 mm). They found relatively low leaching: 5.6 kg N ha$^{-1}$ leached in 2 years when the fertilizer was applied 4 times over the 2-year study period, and 8.6 kg N ha$^{-1}$ leached in 2 years when the fertilizer was applied one time at the beginning of the sampling. Another interesting study on nutrient dynamics in eucalyptus plantation is the one of Laclau et al. 2010 (Forest Ecology and Management 259(9):1771-1785) for plantations of different ages in Brazil and Congo, fertilized once with 38 kg N ha$^{-1}$ (Congo) and 120 kg N ha$^{-1}$ (Brazil). The leaching fluxes were in the range of 1-6 kg N ha$^{-1}$ yr$^{-1}$. Another study by this group, published by Versini et al. 2014 (Geoderma 232-243: 426-436), on 2-years-old eucalyptus plantations in Congo (annual rainfall of 1220 mm and fertilization of 43 kg N ha$^{-1}$ at planting), measured similar leaching losses, equal to 4.4 kg N ha$^{-1}$ yr$^{-1}$. We decided to not include these data on eucalyptus plantations in Table A2 because: 1) the rainfall was much lower than the one at our site 2) the majority of the data were from young plantations, 3) the plantations were not regularly fertilized.

Author's changes in the manuscript: no changes.

L515-517: not sure that the amounts of Na taken up by soil macrofauna could be sufficient to explain the differences. It has been demonstrated that oil palm can take up Na in addition/substitution to K (Bonneau, X., Boutin, D., Bourgoing, R., Sugarianto, J., 1997. Le chlorure de sodium, fertilisant idéal du cocotier en Indonésie. Plantations, Recherche, Développement 4 (5), 336–346), as also shown recently in eucalypt plantations.

Author's response: thank you for providing these interesting references that show that palm and trees in plantations can take up Na from the soil. However, we think that this cannot explain lower Na leaching with mechanical weeding compared to herbicide weeding because it would imply higher Na uptake by the palms with mechanical weeding, which was not investigated.

Author's changes in the manuscript: no changes.

Tables, Figures and appendices The 4 tables, 5 figures and 3 appendices shown are clear and relevant.

Technical corrections

L329: than in the inter-row.

Author's response: corrected.

Author's changes in the manuscript: L 332.

L332: dissolved organic N or total dissolved N?

Author's response: we meant total dissolved N. We understand that this may be misleading and we decided to remove "dissolved".
Author's changes in the manuscript: L 335.

L453: higher?

Author's response: we modified it with "larger".
Author's changes in the manuscript: L 455.

**Comments from Reviewer 3:**

Specific comments

The abstract should be revised.

Line 23-28 Could you consider the specific data (e.g., low solute concentrations, small drainage…) should be added, and thus increasing the persuasiveness for the readers? (Maybe it is important in the Abstract).

Author's response: we agree that is important to insert some specific data but we want to avoid getting a long abstract. We think that the difference in leaching losses between conventional and reduced management are the most important numbers to be shown so we will include them in the abstract.

Author's changes in the manuscript: revisions at L 37-32.

The introduction is very good. The scientific question is clear.

Line 41 The term "e.g" should be "e.g.".

Author's response: corrected.

Author's changes in the manuscript: L 41.

Line 52-54 Could you provide some references? Thank you.

Author's response: since we don´t have strong evidence to justify this sentence, we will restructure it. We provide a reference on the declining in long-term productivity (Syers 1997).

Author's changes in the manuscript: revised sentence at L 50-52.

Line 57-63 Indeed, high precipitation rate is a critical driver for surface runoff and associated nutrient losses. Particularly for considerable plantations. However, the leaching losses may be offset by a high nutrient cycling due to the rapid uptake of plants.

Author's response: thank you for this observation; this was expanded in the manuscript in the second paragraph.

Author's changes in the manuscript: no changes.

Line 92 How far the radius of the palm circle?

Author's response: it is 2 m from the palm circle, this was clearly explained in the materials and method but we will also include in the introduction for more clarity.

Author's changes in the manuscript: we inserted this in L 92.

Line 98-99 Could you describe more details about the root distribution of oil palm? I am not sure the roots of palm only grow around the trunk. I think the root biomass between inter-row area may be high in somewhere.

Author's response: the root density is higher in the palm circle because it´s closer to the palm stem and because of the repetitive fertilization in this area, whereas the inter-row has the lower root density.

Author's changes in the manuscript: a sentence about root density is included at L 97-98.

The Materials and methods section is good structure. The content is detailed and makes it easy for readers to understand.

Line 159 Replace the "x" between "the 50 x 50 m" with "_".

Author's response: we prefer to not modify this because the symbol "_" may cause confusion to the reader since we want to indicate a mathematical product, normally identified by the symbol "x".

Author's changes in the manuscript: no changes.

Line 160 Where is the plant materials from the mechanical weeding? Are they transported far away from the plots?

Author's response: the plant material is left inside the plot.

Author's changes in the manuscript: this was inserted in L 156-157.

Line 232 Is the runoff set to 0? Do you mean "no overland runoff"?

Author's response: yes. For clarity we will insert the term "no overland runoff".

Author's changes in the manuscript: L 353.

Line 243 Soil physical-chemical characteristics.

Author's response: SOC, total N, $^{15}$N natural abundance are the biochemical characteristics, which include the other chemical parameters.

Author's changes in the manuscript: no changes.

Line 247 See comment 1.

Author's response: please see the answer to comment 1.

Line 265 Please simplify the statistical analyses section.

Author's response: although we understand that the statistical analyses section may seem too long, we cannot simplify it without removing essential information. We think that is in line with good scientific practices to report all the statistical analyses used, for reproducibility.

Author's changes in the manuscript: no changes.

The results section is well-organized manner.

However, some statements are so long that they (e.g., the section "3.2 Differences in leaching losses…") should be simplified to delete some non-key contents.

Author's response: we decided included some explanation sentences in the results to guide the reader since we presented a lot of results.

Author's changes in the manuscript: following this comment, the sentence at L 436-439 was revised.

Line 310-311 The drainage flux is low. Do you investigate the stem flow (may be influenced by "funnel effect" of canopy of oil palm?)? Some studies demonstrated that the infiltration was enhanced around the tree trunk.

Author's response: the model used did not allow to parametrize the stem flow and the funnel effect. Nevertheless, the values from the water model were calibrated with field measurements of soil matric potential, reflecting the actual field conditions.

Author's changes in the manuscript: no changes.

Line 346 Why is different between the various elements leaching?

Author's response: the differences in nutrient leaching fluxes between management zones depend on 1) differences in water drainage fluxes (the same of every element) and 2) differences in element concentrations in deep soil-water (which varies for each element). For example, much higher Al concentrations in the inter-row compared to the frond-stacked area resulted in higher Al leaching even if the frond-stacked area have higher water drainage. On the other hand, Mg concentrations were comparable among inter-row and frond-stacked area, but due to the large drainage of the latter Mg leaching was higher in the frond-stacked area.

Author's changes in the manuscript: no changes.

The discussion section was carefully written and prepared.

Line 391 I recommend the ratios of runoff/interception/evaporation/transpiration to precipitation was supplemented in the Table 2 for better understanding.

Author's response: we prefer to not include that in Table 2 to have a simpler table and because the ratios can easily be extrapolated

Author's changes in the manuscript: no changes.

Line 434-438 How to understand the use of organic amendments and slow-release fertilizers? E.g., mulching application? Under the high temperature and precipitation in some tropical areas, the plant materials decompose quickly, and the litterfall may have very short residence time on the ground. Could you provide any information on the standing plant litter in your treatments? Thank you!

Author's response: in the studied plantation, there were two peaks of leaching due to the overlapping of high drainage fluxes and fertilizer application. Since it is complicated to predict the periods with high drainage, it may be useful to use organic or slow-release fertilizers. These can distribute the nutrient input to the soil over a longer period of time, thus reducing the overlapping of high nutrient input and high drainage, and also avoiding peaks of high nutrient inputs. Indeed, the decomposition in the tropics happens fast but it would still provide nutrients slower than mineral fertilization. The organic amendments more used in oil palm plantations are the waste from the palm oil processing, namely the empty fruit bunches (EFB) or the palm oil mill effluent (but this is normally applyied just close to the mill). In a study on decomposition rates and nutrient release by EFB in oil palm plantation, 75% of the EFB was decomposed in 8 months, with a deposition constant k= 0.2 month$^{-1}$ (Moradi et al. 2014, Ann Appl Biol 164: 208–219). Another organic amendment in the oil palm plantation is the litterfall, which is represented by the frond stack. This can provide plenty of nutrients to the plantation as it can be seen from the high nutrient contents in the frond-stacked area in our manuscript (Table 1). Also, the addition of cut fronds is regular, at a rate of 16 fronds palm$^{-1}$ yr$^{-1}$ in the studied plantation. However, this positive effect on soil fertility is restricted to the frond-stacked area and it is unsure if the palm can benefit from these nutrients. Recent literature (Rüegg et al., 2019)

found higher root density under the frond-stacked area compared to the inter-row, indicating that the palm may indeed take up the nutrients from the decomposition of the frond stack.

Author's changes in the manuscript: no changes.

Line 552-553 Although the mechanical weeding is sustainable way in ecological view, the farmers were reluctant to adopt due to its money-consuming and labor-consuming. Undoubtedly, mechanical weeding is a promising measure to reduce nutrient leaching.

Author's response: indeed we would expect the farmers to be reluctant to adopt this weeding method because it requires more labor. However, the economic analysis done in Darras et al. (2019) showed that the costs to implement mechanical weeding would be comparable to the ones for herbicide application. In fact, the prolonged use of glyphosate as standard weeding practice has favored woody and resistant understory vegetation that have to be cut periodically with mechanical weeding.

Author's changes in the manuscript: no changes.

[revised manuscript text omitted]

---

## Author Response (AR2)

Dear Dr. Sara Vicca,

Thank you for accepting our revised manuscript with technical corrections.

Based on your suggestion, we thoroughly revised the manuscript and improved the formulations in many places to improve the readability.

Even if the new version has many changes, such changes are just textual improvements and the contents and interpretations are the same as the previous version.

We hope that our revisions are satisfactory.

Sincerely yours,

Greta Formaglio

[revised manuscript text omitted]